

# Key factors for quantitative precipitation nowcasting using ground weather radar data based on deep learning

Daehyeon Han[1*], Yeji Shin[2*], Jungho Im[1], Juhyun Lee[1]

[1]Department of Urban and Environmental Engineering, Ulsan National Institute of Science and Technology, Ulsan, 44919, South Korea

[2]National Institute of Meteorological Sciences, Korea Meteorological Administration, Jeju-do, South Korea

*Correspondence to*: Jungho Im (ersgis@unist.ac.kr)

* The first two authors equally contributed to the paper.

**Abstract**

Quantitative precipitation nowcasting (QPN) can help to reduce the enormous socioeconomic damage caused by extreme weather. The QPN has been a challenging topic due to rapid atmospheric variability. Recent QPN studies have proposed data-driven models using deep learning (DL) and ground weather radar. Previous studies have primarily focused on developing DL models, but other factors for DL-QPN have not been thoroughly investigated. This study examines four critical factors in the DL-QPN, focusing on their effect on forecasting performance. The four key factors were the prediction design (single, recursive, and multiple predictions), deep learning model (U-Net and convolutional long short-term memory; ConvLSTM), input past sequence length (60 and 120 min), and output future sequence length (60 and 120 min). Twelve schemes were designed to measure the effects of each factor using weather radar data from South Korea. A long-term evaluation for 2018-2020 was conducted from an operational perspective, and a summer heavy rainfall event was analyzed to examine the extreme case. In both cases, U-Net demonstrated the best performance in the critical score index (CSI) with a multiple prediction design. U-Net performed better with shorter input sequences (i.e., 60 min) than with longer input sequences (i.e., 120 min), whereas ConvLSTM did not demonstrate a clear CSI difference over the input sequence length. Future sequence length did not have a significant effect on model performance. All the DL-QPN schemes exhibited underestimation and blurry results as the lead time increased. Through sensitivity analysis, U-Net was highly dependent on the most recent input time (i.e., 0 past minutes), whereas ConvLSTM was more sensitive to various time steps. With an explicit comparison of crucial factors, this study can provide a modeling strategy and contingency plan for future DL-QPN using weather radar data.



# 1 Introduction

Short-term precipitation forecasting is an essential topic in weather forecasting, providing crucial information related to socioeconomic effects in daily life. Short-term rainfall forecasting within two hours is generally called quantitative precipitation nowcasting (QPN), which can be of great assistance in preventing damage from severe precipitation over a short

period (Prudden et al., 2020). Despite the critical importance of QPN, it has been a challenging issue for a long time because of the complexity and dynamic characteristics of the atmosphere (Ravuri et al., 2021). Two major approaches to QPN exist numerical weather prediction (NWP) and statistical extrapolation (Prudden et al., 2020). The NWP simulates future atmospheric conditions, such as precipitation, pressure, temperature, and wind vectors, based on physical governing equations and global data assimilation. Even though NWP has been improved over decades with higher prediction skills and denser

spatiotemporal resolution, NWP for QPN still has limitations due to its high computational cost, synoptic scale prediction, and spin-up issues (Yano et al., 2018; Bowler et al., 2006).  For short term forecasting of precipitation, QPN has generally adopted extrapolation of the sequence of weather radar to focus on local rainfall with relatively high estimation accuracy (Wang et al., 2009; Ravuri et al., 2021; Prudden et al., 2020; Ayzel et al., 2020). Generally, QPN extrapolates the precipitation pattern using only radar sequences (Shi et al., 2015; Ravuri et al., 2021), but it can integrate other data sources, such as weather stations,

NWP, and satellite data (Bowler et al., 2006; Haiden et al., 2011; Chung and Yao, 2020).

Weather radar provides real-time distribution of precipitation with high spatial (approximately 0.5-1 km) and temporal (about 5-10 minutes) resolutions. Various extrapolation approaches have been used for QPN from time-series weather radar data. Temporal extrapolation of the radar sequence demonstrated high prediction accuracy for 1-2 hours lead times, but the performance degraded as lead times increased. Several radar extrapolation methods have been developed, including

thunderstorm identification tracking analysis and nowcasting (TITAN), tracking radar echo by correlation (TREC), and the McGill Algorithm for Prediction Nowcasting by Lagrangian Extrapolation (MAPLE) (Dixon and Wiener, 1993; Mecklenburg et al., 2000; Germann and Zawadzki, 2002; Turner et al., 2004; Germann and Zawadzki, 2004). Despite their superior performance within a few hours compared with NWP, there have been limitations in predicting the onset of precipitation (Kim et al., 2021).

Recent advances in deep learning (DL) have altered conventional weather forecasting methods, especially for short-term predictions like QPN. The radar-based QPN can be viewed as a spatiotemporal video prediction that simulates upcoming frames based on past sequences. Some studies used multiple input sources, such as meteorological variables, ground measurements, and NWP data (Adewoyin et al., 2021; Chen and Wang, 2022; Zhang et al., 2021a; Kim et al., 2021), but the majority of studies used only radar precipitation without any additional input sources. Among DL approaches, convolutional

neural networks (CNNs) are widely used for spatial modeling in computer vision and geoscientific fields. Recurrent neural networks (RNNs) are expected to perform well on time-series datasets owing to their architecture, which recursively feeds the output as the following input and handles successive sequence data. Basic RNNs are not designed to consider spatial information, so there have been attempts to use RNNs for precipitation forecasting for each gauge station (Kang et al., 2020).





Shi et al. (2015) suggested convolutional long-short term memory (ConvLSTM) to combine the benefits of CNN and RNN to

improve QPN performance in Hong Kong. ConvLSTM was designed to model spatiotemporal prediction by applying long-short-term memory (LSTM), one of the most popular RNN models, to convolutional CNN operations. Other RNN-based models of the trajectory gated recurrent unit (TrajGRU) and convolutional gated recurrent unit (ConvGRU) were proposed by (Shi et al., 2017). They reported that the deep learning models outperformed the operational models based on real-time optical flow by variational methods for echoes of radar (ROVER) by the Hong Kong observatory. Several studies using ConvLSTM,

TrajGRU, and ConvGRU have demonstrated the superiority of CNN-RNN models over traditional approaches (Franch et al., 2020; Chen et al., 2020; Zhang et al., 2021b; Ravuri et al., 2021). However, some studies have only used CNNs for DL-QPN. The most widely used model is the U-Net (Ronneberger et al., 2015), which has a U-shaped structure with cascaded encoders, decoders, and skip connections. As U-Net can predict upcoming radar precipitation frames with a more straightforward form than RNN-fused models, it has been widely adopted in recent QPN studies employing deep learning (Ayzel et al., 2020;

Agrawal et al., 2019; Samsi et al., 2019; Ko et al., 2022; Kim and Hong, 2021). Ayzel et al. (2020) reported that RNNs are often unstable. In several application domains, CNNs have demonstrated their numerical robustness during training and made more accurate predictions than RNNs (Bai et al., 2018; Gehring et al., 2017). Recent studies have indicated that deep learning has become the predominant method for QPN owing to its superior performance compared to traditional approaches. However, there is still a dearth of exploration of the various considerations for the DL-QPN besides the DL model itself. As most studies

have primarily focused on developing DL models across multiple study areas and datasets, it is difficult to determine how other factors can affect the skill score, even if they are crucial.

Considering this context, this study investigates critical factors that affect a DL-QPN model. Categorizing key factors in the DL-QPN is challenging, as there is a lack of standard agreement or explicit considerations in the literature. Most previous studies have conducted experiments with different schemes without suggesting clear rationales for their choices. After

analyzing various experimental designs used in previous studies, we summarized the following four potential critical factors in the DL-QPN: (1) the prediction design, (2) the DL model type, (3) the input sequence length, and (4) the output future sequence length. Prediction designs determine how a future sequence is generated. These can be divided into three categories: single, recursive, and multiple predictions. As each prediction scheme has different mechanisms, comparing them can help determine the optimal method for the DL-QPN. The DL model has been the most frequently used factor in previous DL-QPN

studies. The two representative types are fully convolutional networks (FCN) and a combination of CNN and RNN. U-Net (Agrawal et al., 2019; Ayzel et al., 2020; Trebing et al., 2021; Kim and Hong, 2021) and ConvLSTM (Shi et al., 2015; Jeong et al., 2021; Xiong et al., 2021) are the most popular models for FCN and CNN-RNN in DL-QPN, respectively. As the DL-QPN is data-driven, the amount of input and output data will likely determine the model performance. However, the data sequence length has rarely been investigated in previous studies. Various time steps from 25 to 180 min were used as input

data to predict future radar precipitation for up to 360 min, with no explicit comparison in the literature. Hence, the input and output sequence lengths were included as critical factors for further in-depth analysis. A detailed explanation of these four key factors is provided in Section 2.





In this study, we compared 12 DL-QPN schemes considering four factors. Experiments were conducted in South Korea using weather radar data from 2011-2020. As the DL-QPN is highly anticipated to mitigate damage from severe weather, heavy

rainfall events in the Korean Peninsula were examined in detail. A sensitivity analysis was conducted to determine which information in the past time steps contributed the most to each model. As DL is generally regarded as a black box, sensitivity analysis can help to determine which past information DL models utilize more in each input sequence.

The remainder of this paper is organized as follows. A detailed explanation of the four key factors and comparison schemes is provided in section 2. Section 3 describes the data and methods used. Section 4 presents the results, and Section 5 discusses

the results. Finally, Section 6 concludes the paper.

## 2 Key factors in DL-QPN

### 2.1 Prediction design

The four critical factors identified from previous DL-QPN studies using weather radar are summarized in Table 1. Three representative approaches can forecast future rainfall sequences with a given $m$ past rainfall sequence, as depicted in Figure 1.

In a single prediction, each model individually predicts each time step. Because of its simplicity and good performance focusing on a single lead time, image-to-image DL-QPN with radar (Chen et al., 2020) and image-to-point QPN with radar (Kim et al., 2020b) adopt this method. A single prediction employs $n$ distinct models for $n$ time steps (Figure 1a). As shown in Figure 1b, recursive prediction only considers the next time step. The predicted output of the first future time step was fed into an input sequence to predict the next time step, and this process was repeated iteratively to forecast longer lead times.

Ayzel et al. (2020) proposed RainNet v1.0, with a recursive approach, using the previous six sequences to forecast up to 12 future sequences with an interval of 5 min.

Because the recursive model only considers the next time step, it is expected to yield accurate predictions. However, the primary disadvantage of this method is the accumulation of errors with increasing lead time because more predicted results with uncertainty are used as input data. In the multiple prediction, a model simultaneously forecasts all future sequences. This

design has been widely adopted for DL-QPN using a weather radar (Kim and Hong, 2021; Ravuri et al., 2021; Shi et al., 2015; Shi et al., 2017; Franch et al., 2020). A multiple prediction model can generate $n$ time steps, as shown in Figure 1c. Because a multiple prediction model is calibrated for various lead times by minimizing the overall loss, its performance may be degraded for a particular lead time.






**Table 1.** Summary of key factors used in previous image-to-image QPN only using radar sequence based on deep learning. B- the loss column stands for 'Balanced,' which blends different weights for rainfall intensity or each input time.

| Reference | Study area Data Source | Four key factors in this study | | | | Others | |
| --- | --- | --- | --- | --- | --- | --- | --- |
| | | Design | Base DL model | Input scenes (in time) | Prediction scenes (in time) | GAN | Loss |
| Shi et al. (2015) | HongKong Hong Kong Observatory 7 (HKO7) | Multiple | ConvLSTM | 5 (30 min.) | 15 (90 min.) | - | Cross entropy |
| Shi et al. (2017) | HongKong HKO-7 | Multiple | TrajGRU | 5 (30 min.) | 20 (120 min.) | - | B-MSE, B-MAE |
| Agrawal et al. (2019) | USA Multi-Radar/Multi-Sensor System (MRMS) | Unknown | U-Net | Unknown | Unknown | - | Cross entropy |
| Ayzel et al. (2020) | Germany Deutscher Wetterdienst (DWD) | Recursive | U-Net | 6 (30 min ) | 1 (5 min) | - | LogCosh |
| Franch et al. (2020) | Italy Trentino-Alto Adige/Südtirol Radar 2019 (TAASRAD19) | Multiple | TrajGRU | 5 (25 min.) | 20 (100 min.) | - | B-MSE, B-MAE |
| Ravuri et al. (2021) | UK RadarNet4 | Multiple | ConvGRU | 22 (110 min.) | 18 (90 min.) | Conditional GAN | Custom spatial and temporal losses |
| Xiong et al. (2021) | Hong Kong HKO-7 | Multiple | ConvLSTM | 5 (30 min.) | 20 (120 min.) | - | MAE, MSE, B-MAE, B-MSE |
| Cuomo and Chandrasekar (2021) | USA Next Generation Weather Radar (NEXRAD) | Multiple | CNN ConvGRU | 16 (80 min.) | 16 (80 min.) | - | LogCosh |
| Trebing et al. (2021) | Netherlands | Single | U-Net | 12 (60 min.) | 1 (30 min.) | - | MSE |



| | | | | | | | |
|---|---|---|---|---|---|---|---|
| | Koninklijk Nederlands Meteorologisch Instituut (KNMI) | | | | | | |
| Jeong et al. (2021) | Korea KMA | Multiple | CNN ConvLSTM | 18 (180 min.) | 18 (180 min.) | - | MSE |
| (Kim and Hong, 2021; Jeong et al., 2021) | Korea KMA | Multiple | U-Net | 60 min. | 240 min. | Conditional GAN | MAE, MSE, B-MSE |
| Ko et al. (2022) | Korea KMA | Single | U-Net | 7 (70 min.) | 6 (360 min.) | - | B-CSI |
| (Liu et al., 2022) | China China Meteorological Data website (CMD) | Multiple | LSTM | 3 (3 hours) | 3 (3 hours) | - | MAE+MSE |

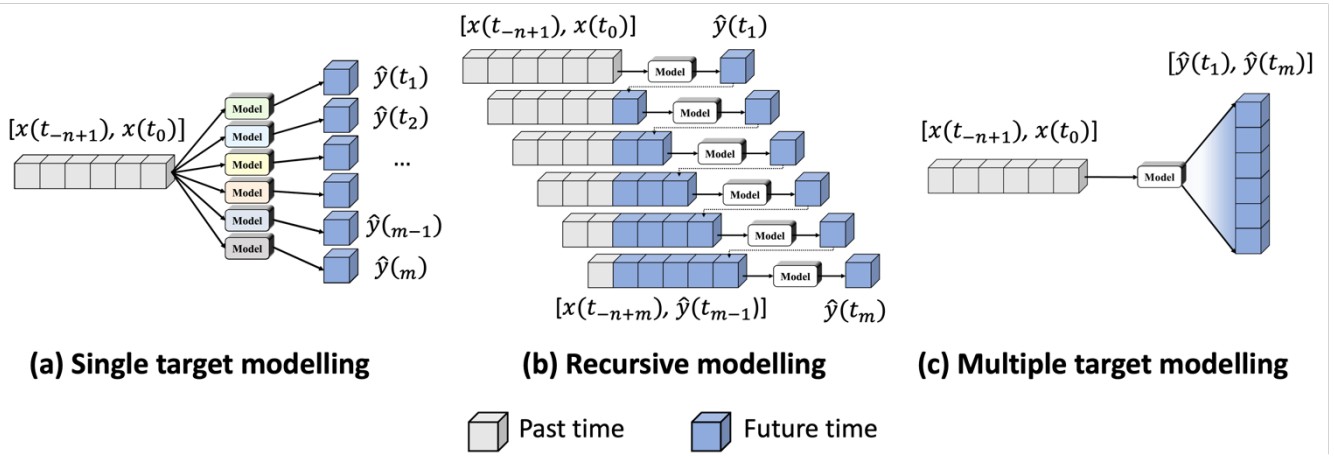

**Figure 1. Three forecasting designs in DL-QPN with radar sequence. (a) Single, (b) recursive, and (c) multiple predictions.**

## 2.2 Deep learning algorithms: U-Net and ConvLSTM

CNNs have been widely adopted in various DL-QPN studies because of their outstanding performance in spatial modeling in various remote sensing and environmental studies including atmosphere (Lee et al., 2021; Gardoll and Boucher, 2022; Geiss et al., 2022; Chattopadhyay et al., 2022), ocean (Chinita et al., 2022; Barth et al., 2022), urban and land (Wu et al., 2022; Sato



and Ise, 2022), and cryosphere (Lu et al., 2022; Kim et al., 2020b). The crucial part of CNNs is finding optimal convolutional filters to predict the target value with input data. The set of convolutional filters with specific window sizes (e.g., $3 \times 3$ or $5 \times 5$) was initialized with dummy values. The initial prediction was generated by conducting the dot product between the filters and the input over the cascading layers. The total error was calculated using the loss function between the model prediction and the actual value. The L1 (mean absolute error (MAE) or L2 (mean squared error (MSE)) loss function is commonly used

in supervised learning for regression; however, other types of loss functions can be used in DL-QPN (Ayzel et al., 2020; Ravuri et al., 2021). Following the loss calculation, the weights of the convolutional layers are updated through backpropagation. With an increasing number of iterations, the model was progressively fitted to the given dataset.

U-Net is the most representative image-to-image model among CNNs. Because DL-QPN can be viewed as image-to-image modeling, U-Net has been widely adopted in recent DL-QPN studies (Ayzel et al., 2020; Bouget et al., 2021; Ko et al., 2022;

Kim and Hong, 2021). U-Net consists solely of convolutional layers that can preserve spatial information from the input to the output. The input for the DL-QPN with weather radar is the previous sequence of the radar images. This model is expected to generate a series of future precipitation scenes. The input and output are image sequences with the dimensions of [*ny, nx, M*] and [*ny, nx, N*], where *M* and *N* are the lengths of the sequence in the past and future, respectively. Its skip connections distinguish U-Net between the same level of encoding and decoding layers, which can mitigate the loss of original information

as the network deepens (Ronneberger et al., 2015). In this study, RainNet v1.0 (https://github.com/hydrogo/rainnet) by Ayzel et al. (2020) was adopted for the U-Net model.

RNNs are expected to yield good performance in time-series forecasting. An RNN is distinguished by its recurrent layers, which feed the output of a specific layer back to its input. As the basic RNN structure suffers from the vanishing gradient problem with an increasing number of recurrent layers, revised RNNs, such as long short term memory (LSTM) and gated

recurrent units (GRU), have gained widespread acceptance (Cho et al., 2014; Hochreiter and Schmidhuber, 1997). They added additional gates to control the information transmitted or dropped, resulting in improved performance compared with vanilla RNNs. Several studies have been conducted on using RNNs for short-term rainfall forecasting (Ni et al., 2020; Srinivas et al., 2019; Aswin et al., 2018). However, these were station-based rainfall predictions that did not account for 2-D information because RNNs were not designed to take spatial information into account. Shi et al. (2015) suggested the Convolutional Long-

Short-Term Memory (ConvLSTM), which combines CNN and LSTM into a single model. Because ConvLSTM has been adopted in recent DL-QPN studies (Chen et al., 2020), it was compared with U-Net in this study. A detailed explanation of the ConvLSTM can be found in Shi et al. (2015).

## 2.3 Length of input radar sequence

The DL-QPN predicts upcoming precipitation based on past sequences. Thus, the composition of the past series directly affects

the model performance. A more extended past sequence may be expected to convey more information than a shorter one; however, the input data distribution may become more diverse as the past sequence lengthens. The optimal length of the past





sequence can vary depending on other factors, such as the forecasting design, DL model, radar time interval, and maximum lead-time. Thus, it is difficult to determine the direct effect of past sequence length on DL-QPN. To test the impact of the input sequence length, radar sequences of 60 and 120 min corresponding to six and 12 radar scenes, respectively, were compared in this study.

## 2.4 Maximum length of future radar sequence for multiple prediction

In addition to the size of the input sequence, the length of the projection is an essential time-related factor. Because the single prediction model is dedicated to each lead time, the maximum size of the future sequence has no effect. In other words, the result of a 30-minute prediction was unaffected when the total lead time for a single prediction was extended from 1 to 2 h. Similarly, as there is only a single model forecasting the next time step in recursive prediction, the maximum length of the future sequence does not affect the results. In the multiple prediction, however, the total lead time does matter, as a model is optimized to fit the overall forecasting skill for each lead time. In the multiple prediction, the forecasting results for a lead time of 30 min are different for the maximum lead times of 1 and 2 h. For this reason, the past sequence length and the total lead time for the multiple prediction should be investigated to determine the effect of input and output length. Hence, only the multiple forecasting schemes with future sequences of 60 and 120 min were compared for the maximum lead time.

## 3. Data and Methods

Figure 2 shows the overall flow of this study. The experiment consisted of three parts: (1) preprocessing, (2) scheme selection, and (3) evaluation. Sections 3.1-3.3 are designed in three parts, consistent with Figure 2.

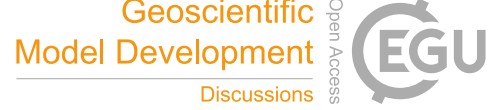

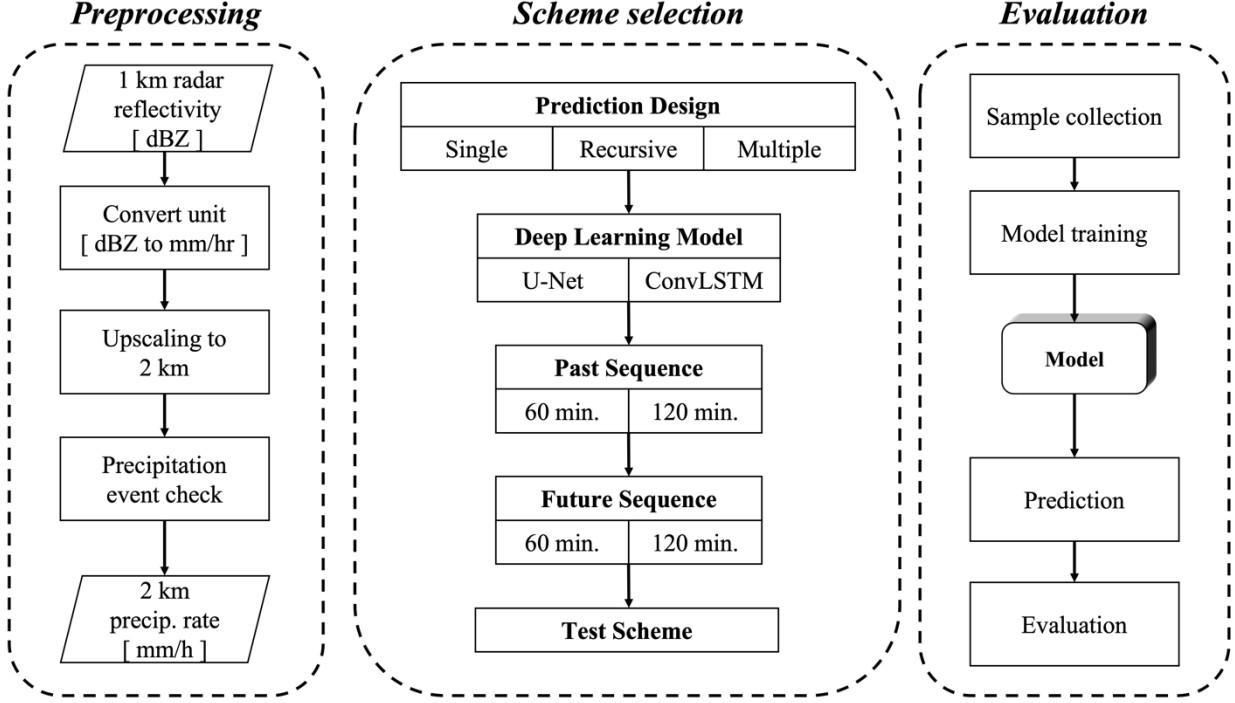

**Figure 2. The proposed overall flowchart to investigate four factors in this study.**

## 3.1 Data and preprocessing

South Korea is in Northeast Asia, with a population of approximately 50 million and numerous industrial facilities. Flooding, inundation, and landslide events occur yearly, especially during the summer monsoon season. Inundation by sudden heavy rainfall can sometimes shut down urban transport systems. Persistent heavy rain during the rainy season (in Korean, Jangma) can cause dams to fail, resulting in massive floods in river basins. Eleven ground weather radars have been operated by the Korea Meteorological Administration (KMA) to monitor precipitation over South Korea. Multiple radars were combined to produce a composite radar reflectance image with a spatial resolution of 1 km (Figure 3). We used the constant altitude plan position indicator (CAPPI), widely used to study precipitation, with an altitude of 1.5 km provided by the KMA (Shi et al., 2017; Han et al., 2019; Kim et al., 2021). To exclude the area outside the radar coverage, we cropped the data from longitude 124E to 131.15E and latitude 33N to 39N. This encompasses most of the national territory of South Korea and a portion of North Korea. For the long-term test, the CAPPI dataset was collected every 10 minutes from 2011 to 2020. A large radar sequence dataset was created using 144 radar scenes daily, with more than 520,000 radar scenes in the study period. Data from 2011-2016 were used for model training, data from 2017 were used for model validation (i.e., hyperparameter optimization), and data from 2018-2020 for the model test. To evaluate the long-term performance from an operational standpoint, we assessed all seasons for three years, 2018-2020, corresponding to approximately 155,520 radar scenes.



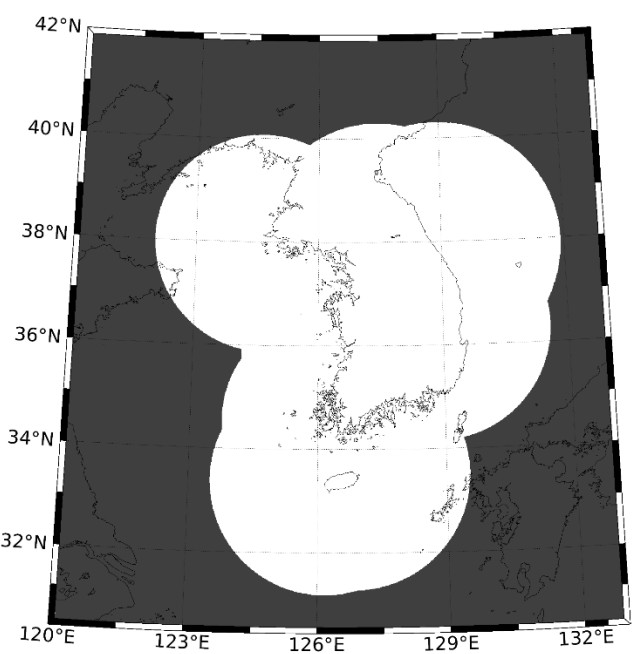

**Figure 3. Weather radar over the Korean peninsula used in this study.**

The KMA weather radar data were provided as reflectance values in decibels relative to Z (dBZ). Marshall–Palmer's Z-R equation (Marshall and Palmer, 1948) converts radar reflectance into precipitation intensity (Equation 1).

$$R = \frac{1}{200} Z^{0.625}, \tag{1}$$

where Z represents reflectance, and R is the precipitation intensity in mm/h. While the original CAPPI radar data had a spatial resolution of 1 km, they were resampled to 2 km, resulting in a 400 × 400 grid to reduce the model training time. As most

pixels in the radar data have no precipitation value, we only used radar scenes when pixels with precipitation intensity higher than 10 mm/h existed in more than 3% of the study area.

MAPLE was used for visual comparison between the DL-QPN schemes and a traditional model. MAPLE predicts the future movement of precipitation echoes by calculating the optimal movement vector using radar reflectivity observations. Overcoming the limitations of simple linear correlation by employing the Lagrangian extrapolation method, MAPLE can

predict precipitation probability by considering the life cycle and scale of precipitation events (Germann and Zawadzki, 2004). In this study, MAPLE from KMA was used. KMA-MAPLE employs hybrid scan reflectivity (HSR) radar composite data with a 1 km spatial resolution centered on the Korean Peninsula. The prediction lead time is up to 6 hours with intervals of 10





minutes. It should be noted that KMA-MAPLE itself is not the current operational product of KMA; thus, the MAPLE in this study does not represent the forecasting skill of precipitation by KMA. As the DL-QPN models and KMA-MAPLE used different types of radar data (i.e., CAPPI and HSR), we did not compare them directly for the quantitative evaluation.

## 3.2 Scheme configuration

Considering these four key factors, 12 experimental schemes were designed (Table 2). The selection of the DL model and its tuning are crucial for maximizing forecasting accuracy. U-Net and ConvLSTM, two representative models in the DL-QPN, were compared in this study. U-Net was adopted for single-, recursive-, and multiple-prediction designs. However, ConvLSTM was adopted only for the multiple prediction, as it is generally expected to be used in time-series datasets. The structures of the U-Net and ConvLSTM are summarized in Figure 4. The U-Net model uses five levels of spatial filters to utilize the different scales of hidden features (Figure 4a). At each level, two convolutional layers with a 3×3 kernel were used. The number of convolutional filters varied depending on the depth of the layer. Skip connections concatenated the equal-sized shallow and deep layers. Several tests were conducted for ConvLSTM to determine the optimal number of layers and hidden states for the ConvLSTM layer, considering the model performance and GPU's memory capacity. Four ConvLSTM layers were used in this study with 16, 32, 32, and 16 hidden states per layer (Figure 4b). All models were trained with the MAE loss function and adaptive momentum optimizer (ADAM), widely adopted in deep learning regression models (Kingma and Ba, 2014). Considering the GPU memory, the batch sizes for U-Net and ConvLSTM were set to 16 and 1, respectively. After the convolutional and ConvLSTM layers, the rectified linear unit (ReLU) and leaky ReLU with 0.1 threshold were used respectively as activation functions to model the nonlinearity of the data. Each model was trained with a maximum of 100 epochs, and the model training was terminated when the validation performance did not improve over three iterations.

Table 2. Specifications of 12 schemes designed in this study with their abbreviations. In the abbreviations, S, R, and M represent single, recursive, and multiple prediction designs, respectively. U stands for U-Net, and C stands for ConvLSTM. The following two numbers are the length of past and future sequence times in minutes.

| Abbreviation | Forecasting design | Model | Input past sequence (# of scenes) | Output future sequence (# of scenes) |
|---|---|---|---|---|
| SU-60-120 | Single | U-Net | 60 min. (6) | 120 min. (12) |
| SU-120-120 | Single | U-Net | 120 min. (12) | 120 min. (12) |
| RU-60-120 | Recursive | U-Net | 60 min. (6) | 120 min. (12) |
| RU-120-120 | Recursive | U-Net | 120 min. (12) | 120 min. (12) |
| MU-60-60 | Multiple | U-Net | 60 min. (6) | 60 min. (6) |
| MU-120-60 | Multiple | U-Net | 120 min. (12) | 60 min. (6) |
| MU-60-120 | Multiple | U-Net | 60 min. (6) | 120 min. (12) |





| MU-120-120 | Multiple | U-Net | 120 min. (12) | 120 min. (12) |
|---|---|---|---|---|
| MC-60-60 | Multiple | ConvLSTM | 60 min. (6) | 60 min. (6) |
| MC-120-60 | Multiple | ConvLSTM | 120 min. (12) | 60 min. (6) |
| MC-60-120 | Multiple | ConvLSTM | 60 min. (6) | 120 min. (12) |
| MC-120-120 | Multiple | ConvLSTM | 120 min. (12) | 120 min. (12) |

**Figure 4. The structure of (a) U-Net and (b) ConvLSTM used in this study.**






It should be noted that other DL models and hyperparameters should be considered in addition to these four factors. The common hyperparameters for deep learning include the loss function, batch size, activation function, and optimizer. Because finding the best set of hyperparameters is time-consuming, other combinations of hyperparameters were not considered at this time. Some recent studies have adopted generative adversarial networks (GAN) in DL-QPN (Ravuri et al., 2021; Kim and
Hong, 2021). Although GAN can be expected to be a promising approach in DL-QPN, we did not compare it in this study because it is far beyond our scope owing to its complexity and diversity.

### 3.3 Evaluation

Three metrics were used to evaluate model performance: the critical score index (CSI), mean absolute error (MAE), and mean
bias (Table 3 and Equations 2-4). To exclude clear scenes from the evaluation, we only used scenes in which the number of pixels with precipitation greater than 1 mm/h exceeded 3% of each scene's total number of pixels. A total of 19,801 scenes met this criterion during the evaluation period from 2018 to 2020, and the percentage of precipitation days was 12.73%. We excluded precipitation of less than 1 mm/h to avoid the effects of clear skies and radar noise. An additional threshold of 5 mm/h was used to account for moderate and heavy precipitation events.
Moreover, a temporal analysis was conducted using summer monsoon rainfall events across the Korean Peninsula in August 2020 to examine the model performance for heavy rainfall phenomena. The summer monsoon rainfall in Korea in 2020 lasted 54 days, from 24 June to 16 August (Lee et al., 2020; Mun et al., 2020). More than 66% of the annual average precipitation during this period fell, with significant regional variation. In particular, 400–600 mm of rainfall fell over the southern part of the Korean Peninsula between 7 and 8 August (Lee et al., 2020; Kim et al., 2020a). Therefore, the specific evaluation period
was set up from 02:30 KST on August 7 to 20:00 KST on August 9, including record-breaking rainfall in the southwestern region.

**Table 3. Confusion matrix for given threshold precipitation $p$.**

|                   | Radar      | Prediction |
|-------------------|------------|------------|
| Hit               | $\geq p$   | $\geq p$   |
| Miss              | $\geq p$   | $< p$      |
| False alarm       | $< p$      | $\geq p$   |
| Correct negative  | $< p$      | $< p$      |

$$MAE = \sum_1^n abs(y - \hat{y}),\tag{2}$$

$$mean\ bias = \sum_1^n(y - \hat{y}),\tag{3}$$





where $y$ is the reference value, and $\hat{y}$ represents the predicted value.

$$CSI = \frac{hit}{hit + miss + false\ alarm}\qquad(4)$$

To measure the contribution of each past time, we conducted a sensitivity analysis by iteratively replacing the input radar scenes with a dummy zero value for all time steps. With the value set to zero for the $i^{th}$ past time, depreciated CSI ($dCSI_i$) can be calculated. The performance drop for the $i^{th}$ past time ($PD_i$) can then be defined as the gap between the original CSI and $dCSI_i$ (Equation 4). The degree of performance degradation can indicate the relative importance of a specific past time.

$$PD_i = CSI - dCSI_i, where\ i\ is\ each\ past\ input\ index\ and\ dCSI\ is\ depreciated\ CSI,\qquad(5)$$

## 4 Results

### 4.1 Model evaluation using precipitation events for 2018-2020

Table 4 shows the performance of all schemes for precipitation events for 2018-2020 with lead times of 10, 30, 60, and 120 min and thresholds of 1 and 5 mm/h. Figures 5 and 6 depict the evaluation metrics with lead times. It should be noted that the

recursive model uses the same model of single prediction for the 10-minute lead time to forecast the next time step. Therefore, the forecasting results of the SU and RU schemes were the same for a lead time of 10 min.

**Table 4. Quantitative performance for all rainy radar scenes in 2018-2020 with the thresholds of 1 mm/h and 5 mm/h. Refer to Table 2 for scheme names. Best scores (i.e., highest CSI and lowest MAE) for each lead time are marked in bold.**

| | Threshold 1 mm/h | | | | | | | | | | | |
| --- | --- | --- | --- | --- | --- | --- | --- | --- | --- | --- | --- | --- |
| Lead time | 10 min. | | | 30 min. | | | 60 min. | | | 120 min. | | |
| Model | CSI | MAE | Bias | CSI | MAE | Bias | CSI | MAE | Bias | CSI | MAE | Bias |
| SU-60-120 | **0.69** | 2.02 | -0.61 | 0.52 | 2.39 | -0.37 | 0.45 | 2.42 | -0.83 | 0.32 | **2.37** | -0.31 |
| SU-120-120 | 0.65 | 2.17 | -0.93 | **0.57** | 2.36 | -0.43 | 0.47 | 2.47 | -0.21 | 0.35 | 2.64 | -0.08 |
| RU-60-120 | **0.69** | 2.02 | -0.61 | 0.50 | 2.45 | -1.28 | 0.32 | 2.71 | -1.54 | 0.19 | 2.64 | -1.19 |
| RU-120-120 | 0.65 | 2.17 | -0.93 | 0.41 | 2.80 | -2.21 | 0.20 | 3.48 | -3.19 | 0.12 | 3.73 | -3.05 |
| MU-60-60 | 0.67 | 2.14 | -0.82 | 0.56 | 2.40 | -1.09 | **0.49** | 2.46 | -0.98 | - | - | - |
| MU-120-60 | 0.66 | 2.13 | -0.13 | 0.56 | **2.32** | -0.84 | 0.45 | 2.60 | -1.17 | - | - | - |
| MU-60-120 | 0.67 | 2.21 | -0.70 | 0.56 | 2.43 | -1.01 | 0.47 | 2.56 | -1.24 | **0.38** | 2.68 | -1.44 |
| MU-120-120 | 0.66 | 2.15 | -0.79 | 0.52 | 2.54 | -1.39 | 0.46 | **2.29** | -1.08 | 0.36 | 2.57 | -1.61 |




| MC-60-60 | 0.66 | 2.05 | -1.10 | 0.50 | 2.56 | -1.51 | 0.37 | 2.91 | -1.90 | - | - | - |
|---|---|---|---|---|---|---|---|---|---|---|---|---|
| MC-120-60 | 0.68 | **2.01** | -1.02 | 0.51 | 2.55 | -1.51 | 0.37 | 2.87 | -1.97 | - | - | - |
| MC-60-120 | 0.64 | 2.14 | -1.34 | 0.48 | 2.62 | -1.73 | 0.36 | 2.95 | -2.17 | 0.24 | 3.32 | -2.80 |
| MC-120-120 | 0.68 | 2.05 | -0.81 | 0.51 | 2.58 | -1.38 | 0.37 | 2.90 | -1.96 | 0.23 | 3.33 | -2.75 |
| Persistence | 0.64 | 2.23 | -0.01 | 0.49 | 3.02 | -0.03 | 0.40 | 3.44 | -0.10 | 0.31 | 3.82 | -0.29 |
| Threshold 5 mm/h | | | | | | | | | | | | |
| Lead time | 10 min. | | | 30 min. | | | 60 min. | | | 120 min. | | |
| Model | CSI | MAE | Bias | CSI | MAE | Bias | CSI | MAE | Bias | CSI | MAE | Bias |
| SU-60-120 | **0.56** | 5.06 | -2.47 | 0.31 | 6.66 | -3.74 | 0.19 | 7.68 | -5.79 | 0.01 | 9.56 | -9.38 |
| SU-120-120 | 0.52 | 5.42 | -3.19 | **0.40** | **6.08** | -2.89 | **0.28** | **6.66** | -3.36 | 0.00 | 9.02 | -8.75 |
| RU-60-120 | **0.56** | 5.06 | -2.47 | 0.28 | 7.47 | -5.48 | 0.11 | 9.22 | -8.09 | 0.03 | 10.28 | -9.67 |
| RU-120-120 | 0.52 | 5.42 | -3.19 | 0.20 | 8.33 | -7.34 | 0.04 | 10.4 | -10.1 | 0.02 | 10.68 | -9.75 |
| MU-60-60 | 0.54 | 5.26 | -3.06 | 0.36 | 6.58 | -4.59 | 0.23 | 7.37 | -5.50 | - | - | - |
| MU-120-60 | **0.56** | **4.96** | -1.40 | 0.34 | 6.53 | -4.35 | 0.21 | 7.58 | -5.32 | **-** | - | - |
| MU-60-120 | 0.55 | 5.12 | -2.41 | 0.36 | 6.48 | -4.23 | 0.22 | 7.40 | -5.77 | 0.09 | **8.65** | -7.77 |
| MU-120-120 | 0.52 | 5.29 | -2.95 | 0.29 | 7.17 | -5.10 | 0.11 | 8.49 | -7.60 | 0.02 | 9.73 | -9.35 |
| MC-60-60 | 0.49 | 5.72 | -4.16 | 0.27 | 7.39 | -5.63 | 0.13 | 8.62 | -7.32 | - | - | - |
| MC-120-60 | 0.50 | 5.72 | -4.26 | 0.26 | 7.49 | -5.78 | 0.12 | 8.77 | -7.72 | - | - | - |
| MC-60-120 | 0.46 | 6.47 | -5.62 | 0.24 | 7.88 | -6.99 | 0.07 | 9.37 | -9.11 | 0.00 | 10.66 | -10.7 |
| MC-120-120 | 0.52 | 5.50 | -3.33 | 0.29 | 7.23 | -5.09 | 0.13 | 8.64 | -7.48 | 0.00 | 10.59 | -10.6 |
| Persistence | 0.48 | 6.26 | 0.01 | 0.31 | 8.28 | -0.02 | 0.22 | 9.10 | -0.13 | **0.15** | 9.73 | -0.48 |




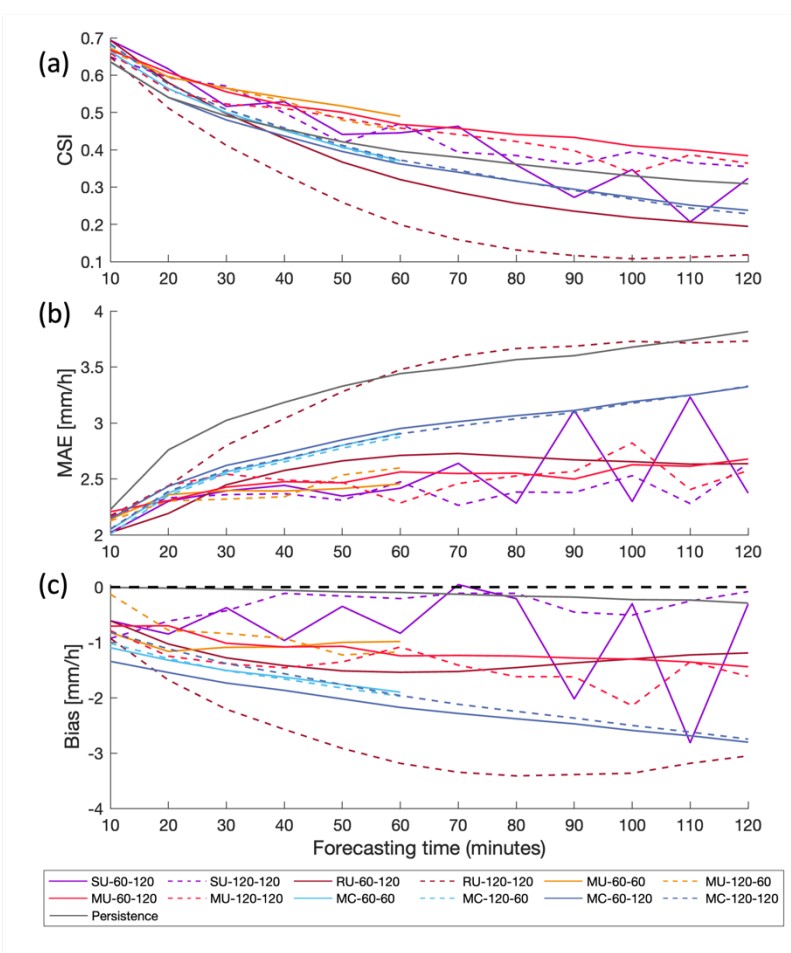

**Figure 5. Performance evaluations of (a) CSI, (b) MAE, and (c) bias over 12 schemes with the threshold of 1 mm/h. Refer to Table 2 for scheme names. A black bold horizontal dashed line in (c) indicates zero bias.**




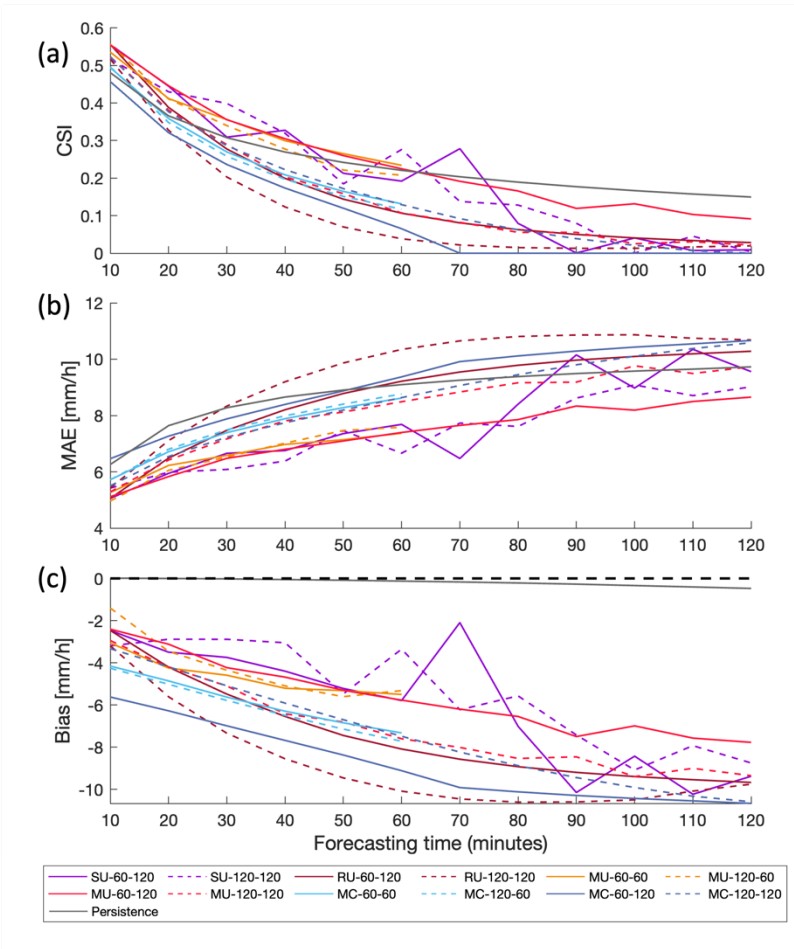

**Figure 6. Performance evaluations of (a) CSI, (b) MAE, and (c) bias over 12 schemes with the threshold of 5 mm/h. Refer to Table 2 for scheme names. A black bold horizontal dashed line in (c) indicates zero bias.**

As noted in Section 3.2, three prediction designs (SU, RU, and MU) were compared for U-Net. With a rainfall threshold of 1 mm/h, SU-60-120 and RU-60-120 showed the highest CSI after 10 min of forecasting (Table 4). The two models produced identical 10-minute forecasts, implying that focusing on a single target for the next step can perform best. However, the overall performance of the three U-Net prediction designs for predicting the next scene was not significantly different. For a lead time of 30 min, SU-120-120 demonstrated the highest CSI score of 0.57, closely followed by the other MU models (Table 4 and Figure 5a). However, when the forecasting time increased to 60 min, MU-60-60 resulted in the highest CSI, followed by other SU and MU models. The SU and MU schemes demonstrated comparable CSI and MAE for all lead times, each achieving the best score with a 1 mm/h threshold (Figures 5 and 6). The CSI performance of the recursive models (RU-60-120 and RU-120-120) was significantly degraded compared with other schemes as the forecasting time increased with the 1 mm/h threshold (Figure 5a). However, this performance gap was reduced to 5 mm/h (Figure 6a). Recursive prediction failed to predict the area



of precipitation, showing a significantly worse CSI score than the SU and MU schemes. Except for some lead times of SU-60-
120, the SU and MU schemes established higher CSI than persistence for all lead times with the 1 mm/h threshold (Figure 5a).
The MAE of persistence with a lead time of 10 min was the highest and increased with longer lead times. The MAE gap
widened as the MAEs of SU and MU became nearly identical over the lead time (Figure 5b). For the mean bias (Figure 5c),
the persistence model showed slight negative tendencies with a minimum value of approximately -0.29 mm/h. In general, the
magnitudes of the negative bias for SU, RU, and MU were greater than those of persistence. This negative mean bias could
indicate an underestimated prediction of the DL-QPN models. Figure 6 shows this underestimation pattern more clearly, as
the CSI and MAE of many DL-QPN schemes became worse than those of persistence with a 5 mm/h threshold. Notably, the
magnitude of the negative bias was significantly larger than 1 mm/h. Even with lead times whose CSI was higher than
persistence (e.g., 10 min), all the DL-QPN schemes resulted in distinct negative biases. This implies that the DL-QPN models
yielded significantly weak intensity predictions, even with a more accurate area prediction than persistence, regardless of the
forecasting design. The common weak-intensity problem is discussed in Section 5.2.

Comparing U-Net and ConvLSTM for the multiple prediction design, the CSI scores of the MU and MC schemes are similar
for a lead time of 10 min. However, all MU schemes showed higher CSI scores than MC schemes for lead times longer than
30 min, with a threshold of 1 mm/h (Table 4 and Figure 5a). The MAE exhibited a similar pattern. The MU schemes yielded
lower MAE than MC for lead times longer than 30 min, with a threshold of 1 mm/h. The magnitude of the negative bias for
MU was generally lower than that for MC. With this overall trend, U-Net seems more appropriate than ConvLSTM in this
study, regardless of the length of the past and future sequences. A more negative bias of the MC schemes indicates that
ConvLSTM is more susceptible to underestimation than U-Net. However, it should be noted that further research is needed
focusing on generalization by comparing them to other study areas and experimental designs.

Figures 5 and 6 compare the difference between the lengths of past sequences of 60 min (solid line) and 120 min (dashed line)
to determine the effect of the length of the input past sequence. For a single prediction, it is difficult to identify the apparent
difference in CSI score between SU-60-120 (solid purple line) and SU-120-120 (dashed purple line) before the lead time of
80 min for both thresholds of 1 and 5 mm/h (Figures 5a and 6a). For lead times longer than 80 min, SU-120-120 showed a
higher CSI than SU-60-120 at a threshold of 1 mm/h (Figure 5a). In terms of recursive prediction, a shorter past sequence (RU-
60-120) exhibited significantly higher CSI and lower MAE than a more extended past sequence (RU-120-120) for both 1 and
5 mm/h thresholds (Table 4, Figures 5 and 6). The magnitude of the negative bias was also significantly greater in RU-120-
120, indicating that a more extended input sequence resulted in a severe underestimation. MU and MC exhibited distinct CSI
patterns for the multiple prediction. In the MU schemes, MU-60-60 and MU-60-120 yielded higher CSI values than MU-120-
60 and MU-120-120, respectively, for both the 1 and 5 mm/h thresholds (Figures 5a and 6a). It is noteworthy that the CSI gap
was more extensive at the threshold of 5 mm/h than at 1 mm/h, implying that a more extended input sequence resulted in a
weaker precipitation intensity, which led to underestimation. On the other hand, there was no distinguished superiority between
60 and 120 min of input sequence in MC schemes with the 1 mm/h threshold, regardless of the lead time (Figure 5). For the
threshold of 5 mm/h, only MC-120-120 had a higher CSI and lower MAE and magnitude of negative bias than MC-60-120,





whereas there was no apparent difference between MC-60-60 and MC-120-60 (Figure 6). This implies that, in addition to the input sequence length, there may be a pattern in the relative ratio of the input to output length, which is further discussed in Section 5.1.


There was no clear pattern in the future sequence length for the multiple prediction design in the MU or MC schemes. For MU schemes with a 1 mm/h threshold, the CSI differences between the shorter future sequences (MU-60-60 and MU-120-60) and longer ones (MU-60-120 and MU-120-120) were not significant (Table 4 and Figure 5a). The MC schemes also did not show a clear pattern for future sequence lengths. Compared with the effect of the past sequence, the future sequence length had a

smaller impact on the model performance for the multiple prediction.

## 4.2 Model evaluation over a heavy rainfall event

### 4.2.1 Time series analysis

Table 5 and Figures 7-10 show the schemes' performance during heavy rainfall from the 7[th] to the 9[th] of August 2020. Figures 7 and 8 represent the CSI in the time series for 214 scenes of the severe rainfall case with rainfall intensity thresholds

of 1 and 5 mm/h, respectively. It should be noted that the performance of all models, including persistence, was affected by the rate of rain pixels in each scene (bold, black line) for both the 1 and 5 mm/h thresholds. Consequently, the performance of the QPN should be interpreted as the precipitation rate because the probability of obtaining a correct prediction (i.e., CSI) increases with the number of precipitation events. The MU schemes yielded the highest CSI, except for a lead time of 10 min (Table 5). With increasing lead times, the CSI gap between the MU schemes and others widened, reaching its maximum at

120 min for both 1 and 5 mm/h thresholds (Figures 7 and 8). In particular, the other schemes were significantly degraded for a lead time of 120 min and a threshold of 5 mm/h. Similar to the long-term evaluation in the previous section, this demonstrated two common drawbacks of DL-QPN models: a performance drop with increasing lead time and an underestimation issue. When a more significant threshold (i.e., 5 mm/h) was used, the underestimation of precipitation became more apparent. Among the 12 schemes, only MU-60-120 exhibited a CSI comparable to the persistence model with a 5 mm/h threshold. The MC

schemes yielded lower CSI than MU, and this performance gap became more extensive when the 5 mm/h threshold was used. Consequently, U-Net demonstrated superiority over ConvLSTM for long-term analysis and heavy rainfall events. The mean biases of all DL models were negative, with a 1 mm/h threshold for all lead times (Table 5 and Figure 9). The magnitude of the negative bias increased when a threshold of 5 mm/h was applied (Table 5 and Figures 10). Most DL-QPN models showed similar time-series patterns in CSI (Figures 7 and 8) and mean bias (Figures 9 and 10).


**Table 5. Quantitative performance over a heavy rainfall event from 7[th] and 9[th] August 2020 with the thresholds of 1 mm/h and 5 mm/h. Refer to Table 2 for scheme names. Best scores (highest CSI and lowest MAE) for each lead time are marked in bold.**

| | Threshold 1 mm/h |
|---|---|
| | |





| Lead time | 10 min. | | | 30 min. | | | 60 min. | | | 120 min. | | |
|---|---|---|---|---|---|---|---|---|---|---|---|---|
| Model | CSI | MAE | Bias | CSI | MAE | Bias | CSI | MAE | Bias | CSI | MAE | Bias |
| SU-60-120 | **0.77** | **3.62** | -1.20 | 0.57 | 4.19 | -1.24 | 0.50 | 4.03 | -1.55 | 0.38 | **3.73** | -1.74 |
| SU-120-120 | 0.74 | 3.87 | -1.76 | 0.64 | 4.09 | -0.75 | 0.54 | 4.05 | -0.66 | 0.41 | 3.88 | -1.39 |
| RU-60-120 | **0.77** | **3.62** | -1.20 | 0.59 | 4.01 | -2.73 | 0.38 | 4.65 | -3.58 | 0.21 | 4.30 | -3.05 |
| RU-120-120 | 0.74 | 3.87 | -1.76 | 0.53 | 4.63 | -4.00 | 0.28 | 5.86 | -5.58 | 0.19 | 5.95 | -5.05 |
| MU-60-60 | 0.74 | 4.03 | -1.48 | **0.65** | 4.11 | -1.80 | 0.57 | 4.12 | -1.88 | - | - | - |
| MU-120-60 | 0.71 | 3.92 | -0.28 | 0.63 | **4.00** | -1.51 | 0.55 | 4.29 | -1.94 | **-** | - | - |
| MU-60-120 | 0.74 | 4.00 | -1.07 | 0.64 | 4.10 | -1.56 | **0.58** | 4.13 | -1.88 | **0.49** | 4.16 | -2.28 |
| MU-120-120 | 0.72 | 4.01 | -1.50 | 0.61 | 4.39 | -2.40 | 0.52 | **3.96** | -2.30 | 0.43 | 4.30 | -3.06 |
| MC-60-60 | 0.75 | 3.75 | -2.25 | 0.58 | 4.42 | -2.81 | 0.46 | 4.86 | -3.34 | - | - | - |
| MC-120-60 | **0.77** | 3.74 | -2.25 | 0.59 | 4.40 | -2.84 | 0.47 | 4.75 | -3.36 | - | - | - |
| MC-60-120 | 0.74 | 3.97 | -2.98 | 0.57 | 4.57 | -3.43 | 0.46 | 4.96 | -3.96 | 0.33 | 5.38 | -4.76 |
| MC-120-120 | **0.77** | 3.77 | -1.93 | 0.59 | 4.45 | -2.71 | 0.47 | 4.86 | -3.48 | 0.33 | 5.32 | -4.55 |
| Persistence | 0.71 | 3.95 | 0.01 | 0.56 | 5.38 | 0.02 | 0.47 | 5.98 | 0.04 | 0.39 | 6.42 | 0.07 |
| Threshold 5 mm/h | | | | | | | | | | | | |
| Lead time | 10 min. | | | 30 min. | | | 60 min. | | | 120 min. | | |
| Model | CSI | MAE | Bias | CSI | MAE | Bias | CSI | MAE | Bias | CSI | MAE | Bias |
| SU-60-120 | **0.67** | **5.12** | -2.90 | 0.39 | 7.76 | -4.38 | 0.28 | 8.69 | -6.25 | 0.02 | 11.37 | -11.1 |
| SU-120-120 | 0.63 | 5.64 | -3.89 | **0.52** | **6.57** | -2.99 | **0.40** | 7.33 | -3.45 | 0.01 | 10.90 | -10.6 |
| RU-60-120 | **0.67** | **5.12** | -2.90 | 0.39 | 8.37 | -6.74 | 0.15 | 10.90 | -10.1 | 0.03 | 12.34 | -12.0 |
| RU-120-120 | 0.63 | 5.64 | -3.89 | 0.29 | 9.56 | -8.92 | 0.04 | 12.27 | -12.2 | 0.03 | 12.39 | -11.4 |
| MU-60-60 | 0.64 | 5.64 | -3.55 | 0.49 | 7.20 | -5.09 | 0.37 | 8.15 | -6.07 | - | - | - |
| MU-120-60 | 0.65 | 5.14 | -1.59 | 0.47 | 7.20 | -4.79 | 0.32 | 8.48 | -5.73 | - | - | - |
| MU-60-120 | 0.66 | 5.42 | -2.75 | 0.50 | 7.02 | -4.76 | 0.37 | **8.06** | -6.17 | 0.20 | **9.58** | -8.47 |
| MU-120-120 | 0.62 | 5.82 | -3.50 | 0.40 | 8.06 | -5.74 | 0.19 | 9.73 | -8.61 | 0.05 | 11.26 | -10.7 |
| MC-60-60 | 0.58 | 6.65 | -5.10 | 0.36 | 8.51 | -6.48 | 0.20 | 9.82 | -8.13 | - | - | - |
| MC-120-60 | 0.59 | 6.64 | -5.21 | 0.35 | 8.61 | -6.67 | 0.19 | 9.96 | -8.55 | - | - | - |
| MC-60-120 | 0.54 | 7.79 | -7.08 | 0.31 | 9.24 | -8.27 | 0.11 | 10.87 | -10.5 | 0.00 | 12.46 | -12.5 |
| MC-120-120 | 0.61 | 6.43 | -4.32 | 0.37 | 8.37 | -6.06 | 0.20 | 9.90 | -8.46 | 0.00 | 12.27 | -12.3 |
| Persistence | 0.55 | 7.28 | 0.05 | 0.37 | 9.69 | 0.08 | 0.28 | 10.66 | 0.13 | **0.21** | 11.34 | 0.21 |




**Figure 7. Comparison of CSI scores for the case of heavy rainfall over South Korea from 7th to 9th August 2020 with the 1 mm/h threshold. Refer to Table 2 for scheme names. The bold black line represents the ratio of precipitation pixels > 1 mm/h for each radar scene.**






**Figure 8. Comparison of CSI scores for the case of heavy rainfall over South Korea from 7th to 9th August 2020 with the 5 mm/h threshold. Refer to Table 2 for scheme names. The bold black line represents the ratio of precipitation pixels > 1 mm/h for each radar scene.**








**Figure 9. Comparison of mean bias for the case of heavy rainfall over South Korea from 7th to 9th August 2020 with the 1 mm/h threshold. Refer to Table 2 for scheme names. The bold black line represents the ratio of precipitation pixels > 1 mm/h for each radar scene.**



Figure 10. Comparison of mean bias for the case of heavy rainfall over South Korea from 7th to 9th August 2020 with the 5 mm/h threshold. Refer to Table 2 for scheme names. The bold black line represents the ratio of precipitation pixels > 1 mm/h for each radar scene.



### 4.2.2 Qualitative comparison with prediction maps

The comparison maps for 05:00, 8th August 2020, with lead times of 10, 30, 60, and 120 min, are depicted in Figures 11-14,
respectively. All U-Net models (SU, RU, and MU) showed a similar pattern, whereas the MC models consistently underestimated peak precipitation, even at a lead time of 10 min (Figure 11). With a lead time of 30 min, the precipitation formation predicted by all DL models started to dissipate, revealing a smoother pattern and weaker intensity (Figure 12). The RU models began to fragment after a lead time of 30 min, and this trend became more apparent with longer lead times (Figures 12c-14c and 12d-14d). In recursive prediction, the degree of deformation and its progression over time varied with the input
sequence length. With a more extended input sequence (RU-120-120), the precipitation area decreased significantly, with more fragmented parcels resulting in a low CSI score. MAPLE consistently maintained peak and overall intensities regardless of the lead time (Figures 11m-14m). In contrast, DL models exhibited a significant decrease in intensity with lead times of 60 and 120 min (Figures 13 and 14). Moreover, the formation of the precipitation area did not disappear and evolved with its tendency in MAPLE. It should be noted that it is hard to compare MAPLE and other DL models and CAPPI ground truth directly due
to the difference in data types (i.e., HSR and CAPPI). With increasing lead times, however, there were significant differences in precipitation area within MAPLE predictions which can indicate that MAPLE can suffer from a low ability to forecast the precipitation area (Figures 11m-14m). Except for the RU and MC schemes, there was less variation in the precipitation area over the lead time in the DL models, especially in terms of displacement. This can indicate that some DL models can achieve better performance than MAPLE in terms of precipitation area detection but fail to predict the growth of intensity and the
details of precipitation development in longer lead times.







**Figure 11. The comparison map in 05:00 8$^{th}$ Aug 2020. in KST with a 10-minute lead time. Refer to Table 2 for scheme names.**

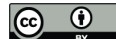



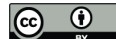


**Figure 12. The comparison map in 05:00 8ᵗʰ Aug 2020. in KST with a 30-minute lead time. Refer to Table 2 for scheme names.**



**Figure 13. The comparison map in 05:00 8<sup>th</sup> Aug 2020. in KST with a 60-minute lead time. Refer to Table 2 for scheme names.**








**Figure 14. The comparison map in 05:00 8$^{th}$ Aug 2020. in KST with a 120-minute lead time. Refer to Table 2 for scheme names.**





### 4.2.3 Sensitivity analysis for the heavy rainfall event

The sensitivity analysis results for the heavy rainfall events are summarized in Tables 6 and 7. In Table 6a, massive
performance drops occurred over most schemes when the latest input information (i.e., 0 past minutes) was replaced with a
dummy value. The drop rate for each scheme ranges from 35% for MC-60-120 to 99% for MU-60-120. Remarkably, the MC
schemes also exhibited a performance decrease for the subsequent input steps (i.e., 10 or 20 min of past time). In contrast, the
MU schemes were hardly affected by other input times at all lead times. This pattern became more evident at a higher threshold
of 5 mm/h (Table 7). This suggests that the U-Net model relied heavily on the last time step of the input sequence, whereas
ConvLSTM utilized information from all time steps more evenly. The difference between the two models may be because of
recurrent modules. Because ConvLSTM is widely known as a time-series model, it may have more room to utilize the
information from various time steps than fully CNN-oriented models, such as U-Net. However, further investigation should
be conducted on this topic because MU generally achieved better performance than MC in this study, indicating that very
recent information may suffice for the DL-QPN. It should be noted that the lower performance drops for longer lead times and
higher thresholds (e.g., forecasting for 120 min with the 5 mm/h threshold in Table 7d) were partially caused by their low
performance. Therefore, it should not be assumed that a lower performance drop indicates greater robustness, given the high-
performance gap between schemes.






**Table 6. Sensitivity analysis for the case study regarding the CSI performance drop with the 1 mm/h threshold. Refer to Table 2 for scheme names.**







**Table 7. Sensitivity analysis for the case study regarding the CSI performance drop with the 5 mm/h threshold. Refer to Table 2 for scheme names.**



## 5 Discussion

### 5.1 Performance comparison and the consideration of key factors


The SU models exhibited varying results for both thresholds (Figures 5 and 6). Because the single prediction design uses *n* separate models for *n* future time steps, it does not guarantee the continuity of the predicted sequence. It is noteworthy that all metrics of SU-60-120 fluctuated more than SU-120-120 for both 1 and 5 mm/h. This suggests that a more extended input sequence could be used to train a more stable set of separated models for each lead time than a shorter sequence in the single

prediction. In recursive prediction, error accumulates as the number of iterations increases. Hence, forecasting as the first step is a crucial part of recursive forecasting. Because the performance of the recursive prediction design significantly degraded as the forecasting time increased (Figures 5 and 6) and the single prediction could not guarantee consistency between predictions;





the multiple prediction scheme would be optimal for DL-QPN with U-Net. In both the long-term evaluation and a heavy rainfall event, U-Net demonstrated a more robust performance in terms of CSI (Tables 4 and 5). This result is consistent with previous studies that reported the superiority of U-Net over ConvLSTM, even with the structural simplicity and fast training time of the U-Net (Ayzel et al., 2020; Ko et al., 2022).

Additional experiments with past and future sequence lengths of 30 and 120 min were added to the multiple prediction design to identify the model performance trend by the ratio of output to input sequences (e.g., the ratio of MU-30-120 is 4, indicating that this scheme forecasts four times longer prediction with the given input data). Tables A1 and A2 summarize the expanded CSI scores of the MU and MC models, respectively. Notably, the MU models performed similarly regardless of the input-output ratio for both intensity thresholds. This extended experiment made it more apparent that U-Net only used information from the most recent few time steps, corresponding to the sensitivity analysis shown in Tables 6 and 7. Moreover, the performance of the MU schemes was hardly affected by future sequence length (Tables A1 and A2). The MC models showed a more diverse distribution of CSI scores over the input-output ratio than the MU schemes, and the variance increased when the 5 mm/h threshold was applied (Tables A1 and A2). However, it is noteworthy that the amount of input information for the past sequence did not significantly affect the model performance, which was unexpected given that the DL-QPN is an entirely data-driven approach with the utmost importance on input data.

## 5.2 Common drawbacks of DL-QPN

In this study, all DL-QPN schemes exhibited a dwindling intensity problem as the lead time increased for both the long-term experiment and the heavy rainfall event. Previous studies have also reported deformation and significant blurry effect of the DL-QPN models (Ayzel et al., 2020; Shi et al., 2015; Trebing et al., 2021). Ravuri et al. (2021) suggested Deep Generative Models of Rainfall (DGMR) to provide realistic rainfall prediction maps. Although DGMR reduced the blurry effect by taking advantage of GAN, it did not perform better than U-Net in terms of CSI (Ko et al., 2022).

Regarding the typical limitations of DL-QPN, the following two factors may play a significant role: (1) the uneven distribution of precipitation and the sparsity of precipitation events and (2) the dynamic movement of the atmosphere. A substantial issue with DL-QPN is the skewed distribution of precipitation towards weak intensities (Shi et al., 2017; Chen and Wang, 2022; Adewoyin et al., 2021). Even after rejecting precipitation events that were too weak (i.e., less than 1 mm/h) during the sampling phase, the pixel-level distribution remained skewed towards the weak range. The sparsity of precipitation events is also strongly related to data imbalances. Because most pixels have no radar signal, the background of the weather radar mainly consists of zero values. This sparsity distinguishes the DL-QPN from general video prediction and makes the model susceptible to the prediction of underestimated results. Data augmentation or patch-level sampling can be used during the sampling phase to reduce data imbalance and sparsity. Advanced loss functions have also been used to solve this skewed distribution during the model training phase (Ravuri et al., 2021; Shi et al., 2017; Ko et al., 2022).





Because the DL-QPN deals with very dynamic atmospheric data, the mismatched position between past and future sequences
can result in significantly degraded performance as the forecasting time increases. The DL-QPN does not explicitly learn the
movement of the precipitation cells. Because convolutional filters in a single layer cannot link remote information beyond the
kernel size, multiple layers gradually extend the receptive field of interest. However, even with multiple layers, the model may
fail to simulate a future precipitation cell whose position is too far from its origin. The Pearson correlation coefficient for heavy
rainfall events was calculated to measure the similarity between the two radar scenes (Table 8). Except for the time gap of 10
min, the correlation coefficients were lower than 0.5 for the 1 mm/h intensity threshold. They fell below 0.05 when the time
difference was extended to approximately 2 h (Table 8a). When only precipitation pixels exceeding 5 mm/h were counted, the
correlation coefficient values became negative with time gaps more significant than 30 min, owing to the movement of
precipitation cells. Weak pixel-level connections between past and future scenes can be highly related to degraded predictions
in terms of both precipitation intensity and formation.

Although recent studies on DL-QPN have reported that they achieved better performance than traditional models with end-to-
end learning, it is crucial to investigate how to exploit precipitation characteristics in DL-QPN fully. The properties of each
precipitation cell can be fed explicitly into the model to successfully simulate the relationship between the precipitation cells
across different time steps. In a similar context, there is ample room to contribute to radar-based DL-QPN using additional
input variables, such as atmospheric instability indices, temperature, vertical humidity profiles, wind vectors, and NWP-
predicted precipitation. Although some previous studies have attempted to fuse heterogeneous datasets for the DL-QPN (Zhang
et al., 2021a; Adewoyin et al., 2021; Bouget et al., 2021), further research is required to determine their contribution and
develop a synergetic model to maximize the multimodal dataset beyond stacking the variables as input channels.

**Table 8. Pearson correlation coefficient for radar precipitation intensity in the heavy rainfall event from 7th to 9th August 2020 over
different past and future times with the thresholds of 1 mm/h and 5 mm/h.**

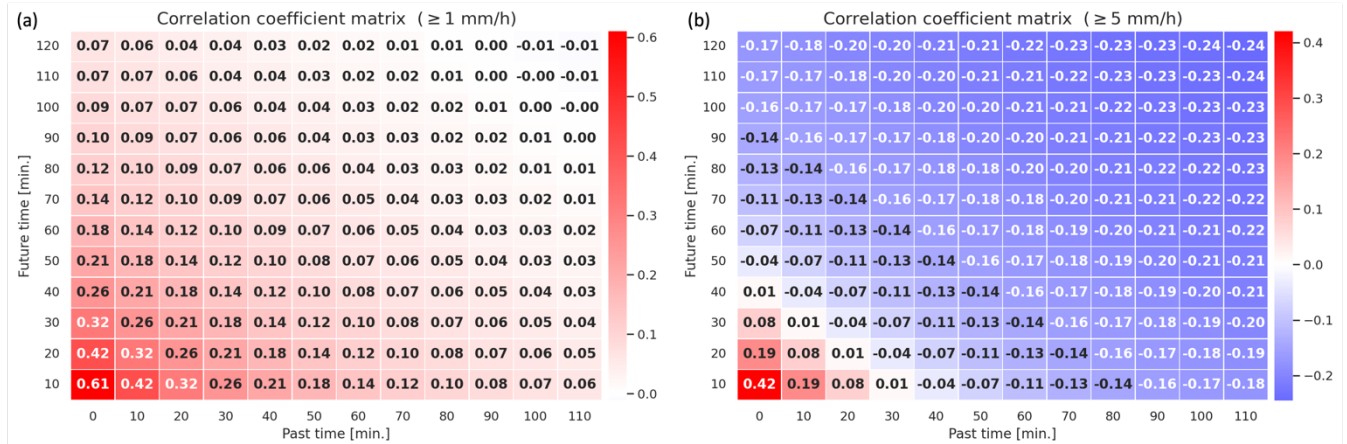





### 5.3 Novelty and Limitations

This study conducted a comprehensive comparison of the DL-QPN. The major novelties of this study are summarized as follows: First, we categorized and investigated four critical factors of DL-QPN. While most previous DL-QPN studies have
focused on DL models, there has been a dearth of research on other factors to be considered. Second, a long-term evaluation was conducted for three years during all four seasons. As most previous studies conducted experiments during rainy seasons, we extended this to the entire year from an operational perspective. Sensitivity analysis of the U-Net and ConvLSTM models revealed the dependence of each model on the input sequence time steps. This demonstrates a clear difference between U-Net and ConvLSTM, which can be used to build the modeling strategy and contingency plan for missing datasets. Finally, we
summarize the common drawbacks of the DL-QPN and discuss their possible causes.

Despite the innovations in investigating key factors in DL-QPN, several limitations remain. One of the significant limitations of this study is the lack of investigation of loss functions. Because the loss function is a guide for model optimization, it is of utmost importance for model training. In addition to L1 and L2 losses, several loss functions have been suggested for DL-QPN, such as logcosh (Ayzel et al., 2020; Cuomo and Chandrasekar, 2021), balanced loss (Shi et al., 2017; Franch et al., 2020;
Xiong et al., 2021; Ko et al., 2022), and adversarial loss using GAN (Ravuri et al., 2021; Kim and Hong, 2021). Although the loss function in the DL-QPN is of the utmost importance, it is beyond the scope of this study, as there are too many aspects to cover in a single paper. Another limitation is the small study area. To generalize the research findings, it would be ideal for examining several study areas with different environments if possible. To the best of our knowledge, most previous studies on DL-QPN using radar have also focused on a single study area, which suffers from the generalization of their proposed model.
Consequently, it is highly expected to evaluate models over multiple study areas in the future to increase operational generalizations. Finally, it should be noted that the model structures used in this study might not be optimal for each scheme because U-Net and ConvLSTM shared the same design across multiple schemes. This may be an inevitable limitation of this study, which was to compare other factors with fixed DL structures.

### 6 Conclusion

This paper summarizes and compares the effects of the four critical factors in the DL-QPN. As previous studies mainly focused on developing DL models with less investigation of other considerations, we expect this study to contribute to future DL-QPN studies by drawing attention to other essential factors. A total of 12 schemes were compared, considering the prediction design, deep learning model, and past and future sequence length. Through quantitative and qualitative comparisons, multiple U-Net predictions appear to be the best combination for this study, considering the CSI skill score and its stable patterns. In general,
the shorter past sequence yielded better performance than the more extended sequence in U-Net. In contrast, it was challenging to find a dominant superiority in the past sequence length in ConvLSTM. According to the sensitivity analysis, U-Net primarily utilizes the most recent input data. ConvLSTM showed a more distributed sensitivity than U-Net, which can be interpreted as

the difference between the fully convolutional model and the RNN-fused network. With extended experiments for multiple prediction schemes, the input-output ratio had less effect than anticipated. Underestimation and smoothed spatial patterns are common drawbacks of the DL-QPN. The skewed distribution of the weak intensity and sparsity of precipitation events are the likely causes of the underestimation, which can be mitigated with improved sampling strategy and loss function in the future.

**Code and data availability**

The original RainNet v1.0 is an open source code provided by Ayzel et al. (2020) via GitHub (https://github.com/hydrogo/rainnet; last access: 11 November 2022). Ground weather radar and MAPLE over South Korea are available at the KMA radar center (http://radar.kma.go.kr; last access: 11 November 2022) and Korea Public Data Portal (https://www.data.go.kr/en/data/15068574/fileData.do, last access: 11 November 2022) upon request. Model structures, trained models, and validation datasets over heavy rainfall events in August 2020 can be found at https://doi.org/10.5281/zenodo.7312779 (Han, 2022).

**Author Contributions**

DH and YS initiated and led this study, conducted experiments and evaluations, and wrote the original article. JI and JL reviewed and edited the manuscript. All authors contributed to designing the methodology. All authors analyzed and discussed the results. JI supervised this study and acquired funding. DH and YS equally contributed to this paper.

**Competing interests**

The contact author has declared that none of the authors has any competing interests.

**Acknowledgement**

We acknowledge KMA for providing the MAPLE and ground weather radar data.

**Financial support**

This study has received funding from "Ministry-Cooperation R&D program of Disaster-Safety" (No. 20009742) by the Ministry of the Interior and Safety (MOIS), South Korea. This research was also supported by "Development of Advanced Science and Technology for Marine Environmental Impact Assessment" of Korea Institute of Marine Science & Technology Promotion (KIMST) funded by the Ministry of Oceans and Fisheries (KIMST-20210427).





**Appendix**

**Table A1. Extended CSI scores for 2018-2020 with a 1 mm/h threshold. The highest score for each lead time is in bold.**

| Model | Threshold 1 mm/h | | | | | | | | | | | |
|---|---|---|---|---|---|---|---|---|---|---|---|---|
| | Lead time (minute) | | | | | | | | | | | |
| | 10 | 20 | 30 | 40 | 50 | 60 | 70 | 80 | 90 | 100 | 110 | 120 |
| MU-30-30 | 0.70 | 0.62 | 0.55 | - | - | - | - | - | - | - | - | - |
| MU-60-30 | 0.70 | 0.60 | 0.54 | - | - | - | - | - | - | - | - | - |
| MU-90-30 | 0.70 | 0.57 | 0.50 | - | - | - | - | - | - | - | - | - |
| MU-120-30 | 0.70 | 0.60 | 0.53 | - | - | - | - | - | - | - | - | - |
| MU-30-60 | **0.72** | **0.63** | 0.56 | 0.52 | 0.48 | 0.45 | - | - | - | - | - | - |
| MU-60-60 | 0.68 | 0.60 | 0.56 | 0.54 | **0.52** | **0.49** | - | - | - | - | - | - |
| MU-90-60 | 0.71 | 0.62 | 0.56 | 0.52 | 0.48 | 0.44 | - | - | - | - | - | - |
| MU-120-60 | 0.67 | 0.60 | 0.56 | 0.53 | 0.48 | 0.46 | - | - | - | - | - | - |
| MU-30-90 | 0.71 | 0.62 | 0.57 | 0.53 | 0.49 | 0.47 | 0.44 | 0.41 | 0.39 | - | - | - |
| MU-60-90 | 0.71 | **0.63** | 0.57 | 0.53 | 0.50 | 0.48 | 0.45 | 0.43 | 0.40 | - | - | - |
| MU-90-90 | 0.70 | 0.60 | 0.55 | 0.50 | 0.48 | 0.46 | 0.44 | 0.40 | 0.37 | - | - | - |
| MU-120-90 | 0.70 | 0.61 | 0.55 | 0.51 | 0.48 | 0.46 | 0.42 | 0.39 | 0.36 | - | - | - |
| MU-30-120 | 0.71 | **0.63** | **0.58** | **0.55** | 0.51 | **0.49** | 0.46 | **0.45** | **0.43** | 0.40 | 0.37 | 0.36 |
| MU-60-120 | 0.67 | 0.61 | 0.55 | 0.52 | 0.50 | 0.47 | **0.46** | 0.44 | **0.43** | **0.41** | **0.40** | **0.38** |
| MU-90-120 | 0.68 | 0.59 | 0.54 | 0.48 | 0.45 | 0.43 | 0.40 | 0.37 | 0.35 | 0.32 | 0.29 | 0.27 |
| MU-120-120 | 0.66 | 0.56 | 0.52 | 0.51 | 0.49 | 0.46 | 0.44 | 0.42 | 0.40 | 0.34 | 0.39 | 0.36 |
| MC-30-30 | 0.67 | 0.57 | 0.48 | - | - | - | - | - | - | - | - | - |
| MC-60-30 | 0.68 | 0.57 | 0.49 | - | - | - | - | - | - | - | - | - |
| MC-90-30 | 0.68 | 0.58 | 0.50 | - | - | - | - | - | - | - | - | - |
| MC-120-30 | 0.69 | 0.56 | 0.48 | - | - | - | - | - | - | - | - | - |
| MC-30-60 | 0.69 | 0.57 | 0.50 | 0.44 | 0.39 | 0.35 | - | - | - | - | - | - |
| MC-60-60 | 0.67 | 0.57 | 0.50 | 0.45 | 0.41 | 0.38 | - | - | - | - | - | - |
| MC-90-60 | 0.67 | 0.58 | 0.52 | 0.47 | 0.44 | 0.41 | - | - | - | - | - | - |
| MC-120-60 | 0.69 | 0.58 | 0.50 | 0.45 | 0.41 | 0.37 | - | - | - | - | - | - |
| MC-30-90 | 0.68 | 0.58 | 0.51 | 0.46 | 0.42 | 0.38 | 0.35 | 0.32 | 0.30 | - | - | - |
| MC-60-90 | 0.66 | 0.58 | 0.52 | 0.47 | 0.43 | 0.40 | 0.37 | 0.34 | 0.32 | - | - | - |
| MC-90-90 | 0.67 | 0.57 | 0.50 | 0.45 | 0.41 | 0.38 | 0.35 | 0.32 | 0.29 | - | - | - |





| MC-120-90 | 0.68 | 0.58 | 0.51 | 0.47 | 0.43 | 0.40 | 0.36 | 0.34 | 0.31 | - | - | - |
|---|---|---|---|---|---|---|---|---|---|---|---|---|
| MC-30-120 | 0.67 | 0.57 | 0.50 | 0.46 | 0.42 | 0.38 | 0.35 | 0.33 | 0.30 | 0.27 | 0.24 | 0.21 |
| MC-60-120 | 0.64 | 0.55 | 0.48 | 0.43 | 0.40 | 0.37 | 0.34 | 0.32 | 0.30 | 0.28 | 0.26 | 0.24 |
| MC-90-120 | 0.67 | 0.56 | 0.49 | 0.44 | 0.40 | 0.37 | 0.33 | 0.31 | 0.28 | 0.25 | 0.23 | 0.21 |
| MC-120-120 | 0.69 | 0.58 | 0.50 | 0.45 | 0.41 | 0.38 | 0.35 | 0.32 | 0.29 | 0.27 | 0.25 | 0.23 |
| Persistence | 0.64 | 0.54 | 0.49 | 0.45 | 0.42 | 0.40 | 0.38 | 0.36 | 0.35 | 0.33 | 0.32 | 0.30 |

565





**Table A2. Extended CSI scores for 2018-2020 with a 5 mm/h threshold. The highest score for each lead time is in bold.**

| Model | \multicolumn Threshold 5 mm/h | | | | | | | | | | | |
|---|---|---|---|---|---|---|---|---|---|---|---|---|
| | Lead time (minute) | | | | | | | | | | | |
| | 10 | 20 | 30 | 40 | 50 | 60 | 70 | 80 | 90 | 100 | 110 | 120 |
| MU-30-30 | 0.55 | 0.42 | 0.34 | - | - | - | - | - | - | - | - | - |
| MU-60-30 | 0.56 | 0.41 | 0.32 | - | - | - | - | - | - | - | - | - |
| MU-90-30 | 0.53 | 0.34 | 0.24 | - | - | - | - | - | - | - | - | - |
| MU-120-30 | **0.57** | 0.42 | 0.33 | - | - | - | - | - | - | - | - | - |
| MU-30-60 | 0.55 | 0.40 | 0.32 | 0.24 | 0.20 | 0.17 | - | - | - | - | - | - |
| MU-60-60 | 0.54 | 0.42 | **0.36** | **0.31** | **0.27** | 0.24 | - | - | - | - | - | - |
| MU-90-60 | **0.57** | 0.44 | **0.36** | 0.30 | 0.24 | 0.20 | - | - | - | - | - | - |
| MU-120-60 | 0.56 | 0.42 | 0.35 | 0.29 | 0.23 | 0.21 | - | - | - | - | - | - |
| MU-30-90 | 0.55 | 0.42 | 0.34 | 0.27 | 0.21 | 0.19 | 0.16 | 0.12 | 0.11 | - | - | - |
| MU-60-90 | 0.56 | 0.44 | **0.36** | 0.29 | 0.25 | 0.21 | 0.18 | 0.14 | 0.12 | - | - | - |
| MU-90-90 | 0.54 | 0.41 | 0.34 | 0.27 | 0.23 | 0.19 | 0.17 | 0.12 | 0.08 | - | - | - |
| MU-120-90 | 0.56 | 0.44 | **0.36** | 0.29 | 0.23 | 0.20 | 0.15 | 0.11 | 0.08 | - | - | - |
| MU-30-120 | **0.57** | 0.44 | 0.35 | 0.29 | 0.23 | 0.20 | 0.16 | 0.13 | 0.10 | 0.09 | 0.06 | 0.05 |
| MU-60-120 | 0.55 | **0.45** | **0.36** | **0.31** | **0.27** | **0.23** | 0.20 | 0.17 | 0.13 | 0.14 | 0.11 | 0.10 |
| MU-90-120 | 0.52 | 0.37 | 0.30 | 0.22 | 0.17 | 0.14 | 0.10 | 0.08 | 0.06 | 0.04 | 0.03 | 0.02 |
| MU-120-120 | 0.53 | 0.39 | 0.30 | 0.21 | 0.16 | 0.11 | 0.08 | 0.06 | 0.06 | 0.03 | 0.03 | 0.02 |
| MC-30-30 | 0.52 | 0.37 | 0.26 | - | - | - | - | - | - | - | - | - |
| MC-60-30 | 0.48 | 0.34 | 0.25 | - | - | - | - | - | - | - | - | - |
| MC-90-30 | 0.48 | 0.34 | 0.25 | - | - | - | - | - | - | - | - | - |
| MC-120-30 | 0.49 | 0.34 | 0.25 | - | - | - | - | - | - | - | - | - |
| MC-30-60 | 0.50 | 0.35 | 0.26 | 0.20 | 0.15 | 0.11 | - | - | - | - | - | - |
| MC-60-60 | 0.50 | 0.36 | 0.28 | 0.21 | 0.17 | 0.14 | - | - | - | - | - | - |
| MC-90-60 | 0.47 | 0.34 | 0.26 | 0.21 | 0.16 | 0.12 | - | - | - | - | - | - |
| MC-120-60 | 0.50 | 0.35 | 0.26 | 0.20 | 0.16 | 0.12 | - | - | - | - | - | - |
| MC-30-90 | 0.46 | 0.33 | 0.25 | 0.19 | 0.14 | 0.11 | 0.08 | 0.05 | 0.03 | - | - | - |
| MC-60-90 | 0.50 | 0.35 | 0.26 | 0.19 | 0.15 | 0.11 | 0.08 | 0.05 | 0.04 | - | - | - |
| MC-90-90 | 0.46 | 0.32 | 0.23 | 0.17 | 0.12 | 0.08 | 0.04 | 0.00 | 0.00 | - | - | - |
| MC-120-90 | 0.44 | 0.31 | 0.22 | 0.16 | 0.12 | 0.08 | 0.03 | 0.00 | 0.00 | - | - | - |
| MC-30-120 | 0.33 | 0.16 | 0.00 | 0.00 | 0.00 | 0.00 | 0.00 | 0.00 | 0.00 | 0.00 | 0.00 | 0.00 |





| | | | | | | | | | | | | |
|---|---|---|---|---|---|---|---|---|---|---|---|---|
| MC-60-120 | 0.46 | 0.33 | 0.24 | 0.18 | 0.12 | 0.07 | 0.00 | 0.00 | 0.00 | 0.00 | 0.00 | 0.00 |
| MC-90-120 | 0.44 | 0.30 | 0.21 | 0.15 | 0.10 | 0.06 | 0.04 | 0.01 | 0.00 | 0.00 | 0.00 | 0.00 |
| MC-120-120 | 0.53 | 0.38 | 0.29 | 0.23 | 0.18 | 0.13 | 0.10 | 0.07 | 0.04 | 0.02 | 0.01 | 0.00 |
| Persistence | 0.49 | 0.37 | 0.31 | 0.27 | 0.25 | 0.22 | **0.21** | **0.19** | **0.18** | **0.17** | **0.16** | **0.15** |

570





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
