# Peer review of "Key factors for quantitative precipitation nowcasting using ground weather radar data based on deep learning"

_Geoscientific Model Development, 2022_

## Referee Comment (RC2)

**Review of "Key factors for quantitative precipitation nowcasting using ground weather radar data based on deep learning"**

This paper compares different ML configurations for precipitation nowcasting. All the ML algorithms compared are deep learning (DL) algorithms. The aim of the paper is to figure out the best configuration and provide some guidelines for the best modeling strategy. I think this goal project is somewhat ill-defined just because there are so many different choices available, and the results of the study are quite inconclusive (discussed below in specific comments); but still it is important to explore these choices systematically as done in the paper.

There are some major issues with the methodology followed in the paper. The DL models trained using the ADAM optimizer can show very different performances based on random initialization – this is what I have seen in the streamflow simulation applications. Therefore, many different models must be trained using different initializations (i.e., different random seeds); an average of the models can be taken as the final model. It does not seem like the authors have followed this procedure; in which case, the results reported are unlikely to be robust. So, this needs to be taken care of before the paper can be considered for publication.

Further, in the multiple prediction case (MU), one expects that the performance will be worse than the single prediction scenario because in the former the model tries to minimize prediction errors at multiple time-steps. But the advantage of using MU design is that it is trying to capture more information from the data and the synergy between the information at different future time-steps helps improve the prediction. I wonder if it is possible to improve the predictions from the MU by adding small layers at the head of the MU where each layer will be dedicated to a single time-step. This can also be seen as the post-processing of MU predictions. This might be helpful in reducing some of the biases that are shown in Table 4.

The authors have compared their results with the 'persistence' prediction. First of all, persistence has not been defined anywhere in the paper. Assuming that it implies that the forecasted value is the same as at the previous time-step, the reported results are not much better than the persistence (both CSI and MAE). Which raises the question if the DL models are actually extracting any significant information from the data? In this regard, why stop at the previous 120 minutes of past data as input? Why not go further in the past; ConvLSTM could extract useful information from a long past sequence.

**Specific comments:**

Line 70: Why are RNNs unstable? Explain.

Line 88: Do you want to say the 'sequence length' and 'forecasting length'? Amount of data is usually reserved for sample size.

Line 131: A 3x3 kernel was used in this study. Why? Were other kernel sizes tried?

Line 148: Vanilla RNNs also have 'exploding gradient' problem.

Line 149: Do you want to say increasing sequence length?

Line 161: Why is more diverse input sequence a problem? Yes, there is tradeoff between the calibration sample size and the input sequence length. But did you test what is the optimal sequence length for your models given the amount of available training data?

Section 3.2: What learning rates were used to train the models?

Equations 2 and 3: Missing $1/n$.

Table 5: Looking at these results, it seems that that there is no clear winner among these models. It really depends upon what we want to achieve (the performance metric used and the time-step in in future where we need the forecast). This is why I say that this project is a bit ambiguous. In fact, the main conclusion seems to be that there cannot be any specific guidelines for developing these models. I suspect the if the authors compute their goodness measures (CSI, MSE, etc.) separately for different region of the South Korea; they might find that different models perform better in different regions. These arguments apply to high rainfall case of Table 7 also.

Figures 5 and 6: Use a different color scheme to differentiate between the models? It is a bit difficult to differentiate between them.

Lines 294-296: These two sentences are not logically connected. Basically, the first part of the second sentence 'The two models produced identical 10-minute forecasts' can be removed.

Line 303: Language typo: gap was reduced with 5 mm/h threshold?

Line 306-307: This sentence is unclear. Rewrite.

Line 312: But even the CSI were not much better than persistence. So, does this sentence really make sense?

Lines 316-324; I am not sure the ConvLSTM has been designed properly. Basically, the sequence length of 12 may not be good enough for convLSTM.

I think it still might be a good idea to include MAPLE results in Tables 5 and 7 even though these are based on different datasets. This would tell us at least something how a physics based model performs in comparison to the DL models.

Lines 403-405: This is not really supported from Figure 11-14.

Lines 427-429: This is not really true. As I have mentioned earlier, ConvLSTM may perform better with larger sequence length.

Lines 447-448: This has not been mentioned earlier. Explain.

---

## Author Comment (AC1)

**Authors' responses (GMD-2022-276)**

The authors would like to thank the editor for your precious time and invaluable comments. The corresponding changes and refinements are highlighted in yellow in the revised paper and are also summarized in our responses below. Authors' responses are in blue. Editor's comments are in black. When the manuscript in cited, it is shown in *italics*.

**Response to RC1**

The manuscript titled 'Key factors for quantitative precipitation nowcasting using ground weather radar data based on deep learning' presented a thorough analysis of different schemes to approach precipitation nowcasting problems using deep-learning techniques. The different schemes were tested using ground weather radar data over South Korea. In recent days, multiple works have explored the application of deep-learning algorithms for quantitative precipitation forecasting. Exploring endless options and schemes is necessary to understand better the feasibility of using these methods in an operational scenario. I appreciate the authors' effort in conducting a systematic analysis and discussions. Some parts of the manuscripts are still hard to understand and not very clear.

➔ The authors are very appreciative of your valuable time and effort in helping us improve our study. Based on your comments, we have updated our manuscript.

Major comments:

1. Data imbalance: The major problem in precipitation nowcasting is the lack of representation of intense precipitation due to data imbalance. Did the authors try to consider this problem in their analysis?

➔ In our original manuscript, data imbalance was not considered, as there were already many factors to compare. However, since data imbalance does matter in precipitation nowcasting, we addressed it in the revised manuscript. To investigate the effect of several trials to cope with data imbalance, we examined balanced loss functions from previous studies (Shi et al., 2017; Franch et al., 2020; Xiong et al., 2021; Kim and Hong). A balanced loss function is an approach that sets different weights for different intensities. Its definition and equation are described in Section 2.3 and Equation 3. Furthermore, we included the histogram of precipitation and corresponding weights for balanced loss in Figure 2 to illustrate the data imbalance and its anticipated impact.

*2.3 Balanced loss function*

**Authors' responses (GMD-2022-276)**

*The loss function guides the direct optimization of DL models. The basic loss function in DL-QPN is MSE. By summing up the error of each pixel, it produces a single value for a given prediction image. As most valid precipitation pixels are severely skewed in weak rainfall intensity (about ≤ 5 mm/h), calculating MSE (Equation 1) with a uniform weight for all pixels might result in an underestimation problem. Shi et al. (2017) suggested the BMSE to mitigate the sample imbalance by using different weights for precipitation intensity (Equation 2).*

$$MSE = \frac{\sum_{i=1}^{N}(y_i - \hat{y}_i)^2}{N} \tag{1}$$

$$BMSE = \frac{\sum_{i=1}^{N} w(y_i)(y_i - \hat{y}_i)^2}{N}, w(y_i) = \begin{cases} 1, & y_i < 2 \\ 2, & 2 < y_i < 5 \\ 5, & 5 < y_i < 10 \\ 10, & 10 < y_i < 30 \\ 30, & y_i > 30 \end{cases} \tag{2}$$

*where y is the reference value, and yˆ represents the predicted value and N is the number of all valid pixels within the radar area. Figure 2 shows the distribution of rainfall intensity and weights for BMSE.*

[Figure]

*Figure 1. Mean distribution of rainfall intensity for the summers of 2020-2022 in a pixel window of 400×400. The blue bar represents the histogram of rainfall intensity. The green line shows the cumulative distribution function. The red line represents the balanced weights for mitigating data imbalances, as suggested by Shi et al. (2017).*

**Authors' responses (GMD-2022-276)**

➔ When comparing the BMSE, it appears to reduce errors in higher intensity compared to the original MSE. However, it also tends to overestimate at low levels, demonstrating that the overall estimation is generally higher than the original. To address the problem caused by data imbalance, we also tested an ensemble of original and Balanced MSEs. The ensemble results exhibit significant improvements in evaluation results, as summarized in Figure 5.

[Figure]

*Figure 5. Quantitative performance over summers of 2020-2022 of lead times of 30 min, 1 h, and 2 h. Please refer to Table 2 for each scheme. The numbers after metrics indicate the thresholds of precipitation for evaluation.*

2. Data: Do the authors consider datasets with overlap when training is done? If t1-tn is

**Authors' responses (GMD-2022-276)**

used as input and tn+1 to tn+m is the forecast, is tn+1 - tn+n used as input for another sample?

➔ Yes, we noticed that there is overlap when the stride of the time step is shorter than the maximum lead time. In our literature review, we found lack of explicit discussion or explanation of how this overlap was handled. However, in response to your comment, we increased the stride time step from 10 minutes to 30 minutes to reduce excessive overlap among samples. Additionally, overlapped data in the same batch might lead to data collinearity and decrease model generalization. We believe that this approach will address the sampling issue and potential problems from data overlap.

3. Equations 2 and 3: The Mean Absolute error and mean bias equation are not normalized. Missing 1/n

➔ Thank you for pointing this out. We have corrected the errors in the Equations.

4. Line 271 and Section 4.2.3: Adding a dummy zero variable to input causes sparsity. Adding white noise is a better idea. But, I feel that the entire part (Section 4.2.3) does not add much to the paper. It just lengthens the paper. I will suggest the authors remove that part.

➔ Several ways exist to check model sensitivity, and our approach might not be the most optimal. Following your suggestion, we removed the sensitivity analysis and focused on other discussions.

5. Table 4 and others: Persistence is not explained previously in the manuscript.

➔ Thank you for your comment. We added it in lines 282-283.

*The n-hour persistence model represents a straightforward approach in which the current precipitation is assumed to persist without any change for the next n hours.*

Minor comments:

Figure 2: Why is dBZ converted to rain rate? Why not just train the model for reflectivity values?

➔ As the final goal of QPN is to determine the amount of precipitation, we used the unit of mm/h. However, forecasting with dBZ is also an active research area. We discuss the options for forecasting using reflectivity or precipitation intensity in Section 5.3.

(Line 459-464)

*As precipitation is calculated from radar reflectivity, direct prediction of the original signal*

*can also be considered. Some previous studies utilized radar reflectivity in DL-QPN (Bonnet et al., 2020; Lepetit et al., 2022; Albu et al., 2022; Han et al., 2022). To our knowledge, there has been few studies comparing radar reflectivity and precipitation intensity directly in DL-QPN. In this study, we chose to forecast precipitation intensity because our final interest is in the strength of the precipitation. However, as the precipitation intensity can be converted from predicted reflectivity, further investigation is needed in the future to find a better skill score.*

Figure 3: It is better to mark the study region on the map.

→ Thank you for your comment. We have updated Figure 3 to only display the study area with valid radar coverage and the position of each radar.

[Figure]

*Figure 2 Weather radar over the Korean Peninsula used in this study. The grey shadow at the boundary indicates the area outside of valid radar coverage. The locations of the eleven weather radars are represented by red dots..*

Line 229: Why is leaky relu not used for U-net and only used for ConvLSTM?

→ As we employed the original models, we retained the model design, including their activation functions.

**Authors' responses (GMD-2022-276)**

Table 2: Why is SU-120-60 or RU-120-60 not considered for analysis? Please justify this in the text.

➔ Since SU-120-120 and RU-120-120 indicate the maximum lead time, they encompass SU-120-60 and RU-120-60, respectively. However, this part was omitted in the revised manuscript, as we excluded the model design from key factors.

Table 4: Is there a reason why the best bias value is not highlighted? Just curious.

➔ Mean bias can signal the overall tendency of underestimation or overestimation. When the magnitude of mean bias is near zero, it may indicate better results if other metrics are similar or improved. However, we cannot assert an optimal mean bias, as substantial positive or negative residuals may result in a zero-like mean bias. We have included a sentence to clarify this.

(Line 301)

*Zero bias does not inherently signify superior performance*

Figure 8 and 10: The thresholds should be 5 mm/h. Please check the captions.

➔ Thank you for your comment. We have updated the time-series figures, which can now be found in Figures 6 and 7 in the revised version.

[Figure]

*Figure 6. Comparison of CSI performance for the case of heavy rainfall over South Korea from 7th to 8th August 2020 with the 1 mm/h threshold. Refer to Table 2 for scheme names. The bottom black line represents the ratio of precipitation pixels > 1 mm/h for each radar scene.*

**Authors' responses (GMD-2022-276)**

[Figure]

***Figure 7. Comparison of CSI performance for the case of heavy rainfall over South Korea from 7th to 8th August 2020 with the 10 mm/h threshold. Refer to Table 2 for scheme names. The bottom black line represents the ratio of precipitation pixels > 10 mm/h for each radar scene.***

---

## Author Comment (AC2)

**Authors' responses (GMD-2022-276)**

The authors would like to thank the editor for your precious time and invaluable comments. The corresponding changes and refinements are highlighted in yellow in the revised paper and are also summarized in our responses below. Authors' responses are in blue. Editor's comments are in black. When the manuscript in cited, it is shown in *italics*.

**Response to RC2**

This paper compares different ML configurations for precipitation nowcasting. All the ML algorithms compared are deep learning (DL) algorithms. The aim of the paper is to figure out the best configuration and provide some guidelines for the best modeling strategy. I think this goal project is somewhat ill-defined just because there are so many different choices available, and the results of the study are quite inconclusive (discussed below in specific comments); but still it is important to explore these choices systematically as done in the paper.

➔ The authors sincerely appreciate your valuable time and effort in providing constructive comments. Based on your suggestions, we have conducted intensive modeling and analysis, which took more time and effort than our original manuscript. By striving to address the suggested comments appropriately, we believe that this paper has been significantly improved in this time, resulting in more robust findings.

There are some major issues with the methodology followed in the paper. The DL models trained using the ADAM optimizer can show very different performances based on random initialization – this is what I have seen in the streamflow simulation applications. Therefore, many different models must be trained using different initializations (i.e., different random seeds); an average of the models can be taken as the final model. It does not seem like the authors have followed this procedure; in which case, the results reported are unlikely to be robust. So, this needs to be taken care of before the paper can be considered for publication.

➔ As you pointed out, the randomness should be considered in DL-QPN to assure robust prediction. We overlooked this in many previous studies. Based on your comment, we ran each scheme five times and aggregated them to generate the final results with random seeds of 0, 999, 2023, 44919, and 2022276. With 12 initial schemes, a total of 60 runs were conducted, which was very time and resource-intensive. Since the number of runs in DL training increased fivefold, we changed the training samples from all seasons to only summers and increased the total period from 2011-2020 to

**Authors' responses (GMD-2022-276)**

2012-2022 by adding newly collected data. This was described in lines 243-257 and Table 2.

(Lines 243-257)

[revised manuscript text omitted]

Further, in the multiple prediction case (MU), one expects that the performance will be worse than the single prediction scenario because in the former the model tries to minimize prediction errors at multiple time-steps. But the advantage of using MU design is that it is trying to capture more information from the data and the synergy between the information at different future time-steps helps improve the prediction. I wonder if it is possible to improve the predictions from the MU by adding small layers at the head of the MU where each layer will be dedicated to a single time-step. This can also be seen as the post-processing of MU predictions. This might be helpful in reducing some of the biases that are shown in Table 4.

➔ When designing the U-Net model for time-series forecasting, the time dimension can generally be treated in the channel axis. In our pilot test, there were several trials to compare the optimal structure among the variants of U-Net. We tested the multi-head structure for different time steps as you suggested, as well as the same structure for each lead time. However, in our preliminary study, we could not find any improvement over the original U-Net structure from RainNet v1.0 (Ayzel et al., 2020). Hence, we adopted the U-Net structure without testing others in our main experiment.

**Authors' responses (GMD-2022-276)**

The authors have compared their results with the 'persistence' prediction. First of all, persistence has not been defined anywhere in the paper. Assuming that it implies that the forecasted value is the same as at the previous time-step, the reported results are not much better than the persistence (both CSI and MAE). Which raises the question if the DL models are actually extracting any significant information from the data? In this regard, why stop at the previous 120 minutes of past data as input? Why not go further in the past; ConvLSTM could extract useful information from a long past sequence.

➔ The missing definition of persistence was also pointed out in RC1. We added its definition in line 282-283.

(Line 282-283)
The *n*-hour persistence model represents a straightforward approach in which the current precipitation is assumed to persist without any change for the next *n* hours.

➔ To test various time steps, we extended the length of the input sequence to up to 3 hours. When the maximum lead time is set at 2 hours, three input sequence lengths (i.e., 1h, 2h, and 3h) can show the effect of shorter, same, and longer input lengths compared to output. The updated input sequence is described in the manuscript. Consequently, the scheme configuration was changed, as shown in Table 2.

(Lines 147-149)
*To examine the impact of the input sequence length, we compared radar sequences of 1 h, 2 h, and 3 h against 12 future radar scenes for 2 h. This sets up a ratio of input-output sequences at 1:2, 1:1, and 3:2.*

➔ In the results, there was a weak tendency of performance improvement with a longer input sequence in terms of MSE and Balanced MSE (BMSE) in both 1h (Table 4) and 2h (Table 5) lead times. The effect of input sequence length was covered in the revised manuscript.

[revised manuscript text omitted]

Specific comments:

Line 70: Why are RNNs unstable? Explain.

➔ In Ayzel et al. (2020), they reported that RNNs are sometimes brittle, and CNNs are more stable, citing Bai et al. (2018) and Gehring et al. (2017). We checked the suggested references and found these supporting parts. The corresponding texts cited references in Ayzel et al. (2020) as follows:

- Bai et al. (2018):
  "However, RNNs have limitations in processing long-term dependencies due to the vanishing gradient problem and are not well-suited for parallelization."
- Gehgring et al. (2017):
  "Compared to recurrent models, computations over all elements can be fully parallelized during training and optimization is easier since the number of non-linearities is fixed and independent of the input length"

➔ As we checked the original references, it seems that they reported the limitations of RNNs as (1) the vanishing gradient problem and (2) difficulty in parallelization. Based on this, we removed this sentence, as it is hard to simply say that "RNN is unstable."

Line 88: Do you want to say the 'sequence length' and 'forecasting length'? Amount of data is usually reserved for sample size.

➔ We updated the sentence to indicate 'sequence length' more clearly.

**Authors' responses (GMD-2022-276)**

(Line 87-88)

*As the DL-QPN is data-driven, the sequence length will likely determine the model performance*

Line 131: A 3x3 kernel was used in this study. Why? Were other kernel sizes tried?

➔ As you pointed out, the size of the kernel can be diverse. However, we set the kernel size to 3x3 for two reasons. (1) As we adopted the U-Net model from RainNet v1.0 (Ayzel et al., 2020), we kept the original kernel size. (2) Prior to Ayzel et al. (2020), Ayzel et al. (2019) explicitly compared the impact of kernel size on the CNN model. They compared kernel sizes of 3x3, 5x5, and 7x7. Contrary to their expectation that increased kernel size may result in better performance, the largest kernel size (7x7) showed the worst result.

(Ayzel et al., 2019)

*"4.3. Kernel size impact*

*Our first guess was that increasing the neural network receptive field by increasing a size of convolutional kernels, would increase the overall network's performance because of accounting of precipitation specific changes from a larger neighborhood. However, the results are different. It is clear that increasing model's receptive field capacity leads to overfitting and increasing uncertainty of predictions. For providing nowcast with the lead time of one hour, we recommend using smaller kernel size of convolutional layers."*

**Authors' responses (GMD-2022-276)**

[Figure]

Fig. 3. Parameters' impact on the verification period performance

Line 148: Vanilla RNNs also have 'exploding gradient' problem.

➜ Here, "basic RNN" indicates a vanilla RNN. To avoid confusion, we changed "basic RNN" to "vanilla RNN" for clarity.

(Line 127)

*As the vanilla RNN structure suffers from the vanishing gradient problem with an increasing number of recurrent hidden layers, revised RNNs, such as long short term memory (LSTM) and gated recurrent units (GRU), have gained widespread acceptance (Cho et al., 2014; Hochreiter and Schmidhuber, 1997).*

Line 149: Do you want to say increasing sequence length?

➜ No. In this sentence, our intention was to discuss the number of hidden layers of RNN regardless of its sequence length. We updated it according to the previous answer.

Line 161: Why is more diverse input sequence a problem? Yes, there is tradeoff between the calibration sample size and the input sequence length. But did you test what is the optimal sequence length for your models given the amount of available training data?

**Authors' responses (GMD-2022-276)**

➡ This sentence is not about the trade-off between sample size and sequence length but simply that a longer sequence can contain both helpful or non-helpful information simultaneously. We updated the sentence for better clarity.

(Lines 144-145)

*A longer past sequence may provide more information than a shorter one, but it could also contain unnecessary information for model training.*

Section 3.2: What learning rates were used to train the models?

➡ The learning rate was set at 0.001 for the ADAM optimizer.

(Lines 221-223)

*All models were trained with the MSE and BMSE loss functions and adaptive momentum optimizer (ADAM) with a learning rate of 0.001, widely adopted in deep learning regression models (Kingma and Ba, 2014).*

Equations 2 and 3: Missing $1/n$.

➡ Thank you for pointing this out. We corrected the errors in Equations 1, 2, and 4. We also added balanced MSE (BMSE) to Equation 2.

(Line 155-159)

$$MSE = \frac{\sum_{i=1}^{N}(y_i - \hat{y}_i)^2}{N} \tag{1}$$

$$BMSE = \frac{\sum_{i=1}^{N} w(y_i)(y_i - \hat{y}_i)^2}{N}, w(y_i) = \begin{cases} 1, & y_i < 2 \\ 2, & 2 < y_i < 5 \\ 5, & 5 < y_i < 10 \\ 10, & 10 < y_i < 30 \\ 30, & y_i > 30 \end{cases} \tag{2}$$

where $y$ is the reference value, and $\hat{y}$ represents the predicted value and $N$ is the number of all valid pixels within the radar area. Figure 2 shows the distribution of rainfall intensity and weights for BMSE.

(Line 276-277)

$$mean\ bias = \frac{\sum_{i=1}^{N}(y_i - \hat{y}_i)}{N} \tag{4}$$

where $y$ is the reference value, and $\hat{y}$ represents the predicted value.

Table 5: Looking at these results, it seems that that there is no clear winner among these models. It really depends upon what we want to achieve (the performance metric used and the timestep in in future where we need the forecast). This is why I say that this project is a bit ambiguous. In fact, the main conclusion seems to be that there cannot be any specific guidelines for developing these models. I suspect the if the authors compute their goodness measures (CSI, MSE, etc.) separately for different region of the South Korea; they might find that different models perform better in different regions. These arguments apply to high rainfall case of Table 7 also.

➔ As you pointed out, the original manuscript's conclusion was ambiguous due to the absence of clear superiority among the compared schemes. In our revised version, we updated the key factors by dropping prediction design and maximum length and newly added the balanced loss function and ensemble approach. These updated factors convey a more significant comparison with conclusive discussion, as well as some enhanced results compared to the original design, as shown in the previous answers. Consequently, we updated the discussion about performance in Section 5.1.

*5.1 Performance comparison and considerations of key factors*

*To address data imbalance and improve skill scores, various loss functions have been considered in previous research. Our comparison of two representative losses, MSE and BMSE, revealed that each has its strengths and weaknesses. The selection of an appropriate loss function should be informed by a comprehensive evaluation of QPN results. Optimal loss functions may vary depending on the specific objectives as it provides guidance to DL modeling. For instance, if the model's focus is on severe weather, BMSE can be weighted to emphasize high intensities. In cases where the area of precipitation over a certain threshold is of key interest, a modified CSI loss can be used (Ko et al., 2022). As a single metric cannot fully evaluate a model, combinations of different losses can also be explored. Alternatively, the ensemble approach analyzed in our study can leverage different loss functions to create a synergistic effect for QPN.*

*In this study, U-Net consistently outperformed ConvLSTM in various respects, both in long-term evaluation and in a single heavy rainfall event. This finding is in line with previous research (Ayzel et al., 2020; Ko et al., 2022; Han et al., 2023). Additionally, U-Net demonstrated more stability across different random seeds than ConvLSTM (Figure 8). Contrary to the widespread expectation that DL models powered by RNN would excel in time-series forecasting, it was found that a model relying solely on CNN can perform better. However, this does not imply that all models using RNN structures are inferior to full CNN*

**Authors' responses (GMD-2022-276)**

*models. Considering the wide range of U-Net and ConvLSTM variants, there could be potential for RNN-powered models to exhibit superior results. Lastly, the input sequence length did not significantly impact the results compared to other factors in this study. Nevertheless, sequence length should still be carefully considered as DL-QPN relies significantly on past information. In other DL models and QPN designs, input sequence length may have a greater impact than it did in this study, therefore, we continue to regard this as a key factor in DL-QPN.*

*Due to the inherent randomness and stochastic nature of deep learning, modeling and evaluation need to be carefully conducted, taking into account relevant factors. As demonstrated in Figure 8, results can vary for each run with a different random seed. Thus, stability should be a priority when developing a DL-QPN model, a point often overlooked in previous studies. By treating each run as an ensemble member, we can avoid unstable results under varying conditions of randomness.*

Figures 5 and 6: Use a different color scheme to differentiate between the models? It is a bit difficult to differentiate between them.

➔ We believe that evaluation at every time step is not necessary. Therefore, we removed the time-series figures for 10-120 minutes. We have updated the figures for the time-series of heavy rainfall cases in the revised manuscript for better readability.

[Figure]

*Figure 6. Comparison of CSI performance for the case of heavy rainfall over South Korea from 7th to 8th August 2020 with the 1 mm/h threshold. Refer to Table 2 for scheme names. The bottom black line represents the ratio of precipitation pixels > 1 mm/h for each radar scene.*

[Figure]

*Figure 7. Comparison of CSI performance for the case of heavy rainfall over South Korea from 7th to 8th August 2020 with the 10 mm/h threshold. Refer to Table 2 for scheme names. The bottom black line represents the ratio of precipitation pixels > 10 mm/h for each radar scene.*

Lines 294-296: These two sentences are not logically connected. Basically, the first part of the second sentence 'The two models produced identical 10-minute forecasts' can be removed.

Line 303: Language typo: gap was reduced with 5 mm/h threshold?

Line 306-307: This sentence is unclear. Rewrite.

Line 312: But even the CSI were not much better than persistence. So, does this sentence really make sense?

➔ In the revised version, these parts were removed as the result was updated with a new experimental design.

**Authors' responses (GMD-2022-276)**

Lines 316-324; I am not sure the ConvLSTM has been designed properly. Basically, the sequence length of 12 may not be good enough for convLSTM.

➔ Based on your comment, we conducted experiments again for ConvLSTM, extending the input sequence to up to 3 hours, which is longer than the maximum lead time.

I think it still might be a good idea to include MAPLE results in Tables 5 and 7 even though these are based on different datasets. This would tell us at least something how a physics based model performs in comparison to the DL models.

➔ We agree with you that a comparison with a non-DL model would make this evaluation more meaningful. However, as MAPLE is based on a different data type with different quality control and masking, our concern is that it is hard to compare MAPLE (hybrid surface radar) with other DL models (radar CAPPI). Since MAPLE data was provided by the Korea Meteorological Administration, there was a limitation in changing the input data source. Hence, we replaced MAPLE with pySTEPS, a method widely used to demonstrate how non-DL models perform. By adopting pySTEPS, we can directly compare the results. The quantitative and qualitative comparisons with pySTEPS are provided in Tables 5 and 6, and Figures 6-8. As other results were already cited in the previous answers, we only cite Figure 8 here.

➔ Thank you for your comment. Similar to the previous comment, there was no quantitative evaluation using MAPLE. In lines 403-405, we discussed this based on the visual interpretation of Figures 11-14, which was still unclear. After replacing MAPLE with pySTEPS, we were able to calculate the CSI score, indicating how each model can predict the area over a given threshold. In the revised manuscript, the updated results now support that some DL models can achieve better performance than pySTEPS in terms of forecasting precipitation area.

*3.2 Comparison with non-DL model*
*To compare the DL-QPN results with a non-DL model, we also included pySTEPS (Pulkkinen et al., 2019) in our comparison. PySTEPS is a Python implementation of the Short-Term Ensemble Prediction System (STEPS) proposed by Bowler et al. (2006). It has been widely used as a control non-DL model in previous studies (Ravuri et al., 2021; Choi and Kim, 2022; Han et al., 2023; Zhang et al., 2023). By calculating the mean wind vector using the input*

*radar sequence, pySTEPS simulates future radar sequences. To examine the impact of input sequence length, we also tested 1-3 h of input sequence in pySTEPS to predict a maximum of 2 h, the same as the other DL models. More detailed information and usage of pySTEPS can be found in its documentation (https://pysteps.github.io) and repository (https://github.com/pySTEPS/pysteps).*

[Figure]

**Figure 8. Comparison map for 05:00 on August 8, 2020, in KST with a 2h lead time. Refer to Table 2 for scheme names. Schemes with seed numbers are the averages of results for three scenes using 1h, 2h, and 3h input sequences. CSI1 and CSI10 indicate the CSI scores with thresholds of 1 and 10 mm/h, respectively.**

Lines 403-405: This is not really supported from Figure 11-14.

**Authors' responses (GMD-2022-276)**

➔ Thank you for your comment. Similar to previous comment, there was no quantitative evaluation using MAPLE. In lines 403-405, we discussed based on the visual interpretation of Figures 11-14, which is still not clear. After replacing MAPLE with pySTEPS, we can calculate CSI score which can indicate how each model can predict the area over the given threshold. In the revised manuscript, updated results now can support that some DL models can achieve better performance than pySTEPS in terms of forecasting precipitation area.

Lines 427-429: This is not really true. As I have mentioned earlier, ConvLSTM may perform better with larger sequence length.

➔ As in previous responses, we did recognize the need for testing with a longer input sequence. We extended the length up to 3 hours and found a performance improvement. Even though the performance slightly improved in terms of MSE or BMSE, U-Net still outperformed ConvLSTM in this time. Moreover, ConvLSTM exhibited higher variance depending on different random seeds and loss functions. The input length had little impact on both models.

➔ Based on a comment from RC1, we removed Section 4.2.3 to avoid a lengthy manuscript and to focus on other discussions. Therefore, these sentences were also removed from the revised manuscript.

Lines 447-448: This has not been mentioned earlier. Explain.

➔ This part was also removed to accommodate new results.